# MXene/Carbon Nanocomposites for Water Treatment

**DOI:** 10.3390/membranes14090184

**Published:** 2024-08-25

**Authors:** Aruzhan Keneshbekova, Gaukhar Smagulova, Bayan Kaidar, Aigerim Imash, Akram Ilyanov, Ramazan Kazhdanbekov, Eleonora Yensep, Aidos Lesbayev

**Affiliations:** 1Institute of Combustion Problems, 172 Bogenbay Batyr Str., Almaty 050012, Kazakhstan; a.keneshbekova@icp.kz (A.K.); imash.aigerim@icp.kz (A.I.); 2Department of “General Physics”, Intistute of Energy and Mechanical Engineering Named after A. Burkitbayev, Satbayev University, 22a Satpaev Str., Almaty 050013, Kazakhstan; info@kaznu.edu.kz (A.I.); kazhdanbekov_ramazan@live.kaznu.kz (R.K.); yensep_eleonora2@live.kaznu.kz (E.Y.); a.lesbayev@satbayev.university (A.L.); 3Faculty of Chemistry and Chemical Technology, Al Farabi Kazakh National University, 71 al-Farabi Ave., Almaty 050040, Kazakhstan

**Keywords:** MXene, carbon nanomaterials, water treatment, adsorption, rejection, organic and inorganic pollutants

## Abstract

One of the most critical problems faced by modern civilization is the depletion of freshwater resources due to their continuous consumption and contamination with different organic and inorganic pollutants. This paper considers the potential of already discovered MXenes in combination with carbon nanomaterials to address this problem. MXene appears to be a highly promising candidate for water purification due to its large surface area and electrochemical activity. However, the problems of swelling, stability, high cost, and scalability need to be overcome. The synthesis methods for MXene and its composites with graphene oxide, carbon nanotubes, carbon nanofibers, and cellulose nanofibers, along with their structure, properties, and mechanisms for removing various pollutants from water, are described. This review discusses the synthesis methods, properties, and mechanisms of water purification using MXene and its composites. It also explores the fundamental aspects of MXene/carbon nanocomposites in various forms, such as membranes, aerogels, and textiles. A comparative analysis of the latest research on this topic shows the progress in this field and the limitations for the practical application of MXene/carbon nanocomposites to solve the problem of drinking water scarcity. Consequently, this review demonstrates the relevance and promise of the material and underscores the importance of further research and development of MXene/carbon nanocomposites to provide effective water treatment solutions.

## 1. Introduction

Population growth, industrialization, and the expansion of the industrial and agricultural sectors have led to a significant increase in global water consumption, as well as an annual acceleration in the release of pollutants into water sources [1]. Pollutants such as microplastics, pharmaceuticals, and radioactive elements [2,3,4] pose a serious threat to ecosystems and underscore the urgent need for effective water treatment and access to clean water. For example, Tania Montoto-Martínez et al. [5] report that 80% of all marine debris is plastic, and with the growing plastics industry, the amount of plastic debris could reach 12 billion tons by 2050 [6]. On the other hand, it is predicted that about one third of the world’s countries could face severe water shortages by 2040 (Figure 1) [7].

Common water treatment methods include evaporation [8], distillation [9], sedimentation [10], filtration [11], crystallization [12], coagulation [13], electrochemical treatment [14], ion exchange [15], sorption using carbon nanomaterials [16], ozonation [17], chlorination [18], ultrasonic treatment [19], and membrane technologies [20]. Membrane processes employed in the chemical, pharmaceutical, and water industries offer advantages such as high reliability, energy savings, and reduction of secondary pollution [20,21]. However, optimizing membrane processes requires balancing permeability and selectivity, which has stimulated the development of new membranes incorporating nanomaterials and two-dimensional materials (2DM).

Conventional membranes, which separate particles and molecules by size using exclusion and diffusion mechanisms, are currently mass-produced. They are employed in microfiltration, ultrafiltration, nanofiltration, and reverse osmosis to separate substances based on pore size and pressure gradients [22,23]. Examples of conventional membranes include ceramic membranes made from SiO_2_, Al_2_O_3_, and TiO_2_, which are widely used due to their long lifespan [24], as well as polymeric membranes, which are preferred for their stability [25]. These membranes effectively filter particles and bacteria, but they may struggle with the selective removal of certain contaminants without additional treatment steps [22,26].

Adsorptive membranes are also used, designed to remove contaminants from liquids through adsorption, where molecules adhere to the surface of the membrane material [27]. These membranes often contain activated carbon with a large surface area [28], enhancing their ability to capture contaminants.

However, both types of membranes face limitations in terms of purification efficiency, permeability, stability, and selectivity, leading to an increasing need for modifications and the development of new membranes for water treatment. Modern filtration systems are focusing on combining conventional membranes, which rely on physical size separation, with adsorptive membranes that leverage chemical interactions between the adsorbent and contaminants molecules [29].

On the other hand, since the discovery of graphene in 2004 [30], research on 2DMs such as carbon nitrides, metal-organic frameworks, transition metal dichalcogenides, layered double hydroxides, and zeolites [31,32,33,34] has progressed significantly. The high aspect ratio of 2DMs allows the formation of ultra-thin selective layers on porous filter substrates, improving their efficiency. MXene, first synthesized in 2011 [35], has attracted considerable attention among 2DMs due to its high stability, large surface area, high electrical and thermal conductivity, and hydrophilicity. In addition, MXene membranes combine the characteristics of both traditional and adsorptive membranes, meeting the current requirements for membrane technology. They can be designed for efficient size separation, similar to traditional membranes, while offering enhanced adsorption properties due to their unique surface characteristics [36].

Recent investigations have explored the use of MXene in various applications including batteries [37], catalytic processes [38], sensors [39], radiation shielding materials [40], and membranes for water purification [41] (Figure 2). However, the use of MXene in membranes faces problems of repacking and aggregation of nanosheets due to van der Waals interactions [42], as well as swelling in aqueous solutions, which reduces filtration efficiency [43].

The versatile structure of MXene allows it to be tailored and combined with other materials as metal oxides [44], polymers [45], covalent organic frameworks [46], and carbon nanomaterials [47,48,49,50]. Carbon nanomaterials, such as graphene oxide, carbon nanotubes, and carbon nanofibers, are of particular interest due to their unique adsorption properties and environmental neutrality. Moreover, the adsorption properties of carbon nanomaterials arise from different mechanisms. For graphene oxide, its functional groups (e.g., hydroxyl, epoxide, carboxyl) provide excellent adsorption capabilities. For CNTs and CNFs, this is due to their porous structure and high surface area. Meanwhile, their environmental neutrality is attributed to their durability, utilization of abundant or renewable carbon sources, and non-toxic degradation products [51]. This is why membranes based on these materials interest many scientists, despite the challenges associated with fouling and contamination [52,53,54]. However, the combination of MXene and carbon nanomaterials can create a synergistic effect, overcoming the limitations of each material and improving the efficiency of water purification.

Thus, among the previously mentioned membranes, novel membranes and nanostructures based on MXene and carbon nanomaterials stand out as the most promising and advanced, which will be discussed next. This review evaluates the potential of MXene/carbon nanocomposites for water purification, highlighting their prospects and directions for future research. Correspondingly, in view of the novelty of this approach, the topic is relevant and forward-looking.

## 2. Properties and Synthesis of MXene

In accordance with the literature [55,56,57,58,59], MXenes are mainly synthesized by etching A layers from MAX phases, which are ternary carbides or nitrides with the general chemical formula M_n+1_X_n_, where M indicates an early transition metal such as Ti, V, Nb, Mo; A stands for an element of Group IIIA or IVA; X for C and/or N; and n = 1, 2, 3. The MAX structure is layered hexagonally, with alternating M_n+1_X_n_ and A layers. The A layers can be chemically removed without breaking the M-X bonds because the M-X bonds are stronger than the M-A bonds. Gentle exfoliation of the A layers leads to the formation of weakly bonded M_n+1_X_n_ layers, which can be easily separated into layers by ultrasonic treatment. The 2DMs formed are called MXenes to emphasize the loss of layers from the original MAX phase and their two-dimensional nature, similar to graphene. It should be noted that the use of group 13 or 14 elements such as Al for the A layers in MAX is the most common way of synthesizing MXenes. In addition, during the surface etching process, M_n+1_X_n_ elements always form a coating with functional groups such as oxygen (=O), hydroxyl (-OH), and/or fluorine (-F). Therefore, the chemical formula of MXenes is M_n+1_X_n_T_x_, where T_x_ represents surface functional groups. Research shows that the ratio of different functional groups on the MXene surface can be uncertain and variable depending on the etching process conditions.

Furthermore, when two types of transition metals are present in the structure of MXenes in the M-layers, these materials are called double transition metal-MXenes. When the two types of transition metals are either out of plane or in plane, the resulting products are called ordered double transition metal MXenes [60]. To date, researchers have been able to experimentally synthesize about 30 different types of MXenes out of 100 theoretically available [56]. It is expected that the development of new methods of MXene synthesis, such as the use of polar organic solvents, acidic melts of Lewis salts, and others, will lead to emergence of new varieties [61].

MXenes have a number of attractive properties due to their unique two-dimensional structures and the ability to tune the surface chemistry. These include electrical conductivity [62], high specific surface area [63], high strain energy at fracture [64], excellent Young’s modulus [65], and the ability to tune magnetic properties as a function of strain and temperature [66]. In addition, MXene is highly hydrophilic, which improves its interaction with water and contaminants [67]. The surface of MXene can be modified with different functional groups, allowing the adsorption properties to be tailored to specific types of pollutants. Numerous hydrophilic functional groups help to significantly improve the performance of MXene-based membranes [68]. Huang Z. et al. [69] highlight the optical properties of MXene, which are important for membranes with photocatalytic properties or photothermal conversion abilities. MXenes can absorb visible light, which enables effective capture of solar radiation and photothermal conversion into ultraviolet and near-infrared regions. Additional properties of MXenes, such as the presence of a large number of oxygen groups in the basal plane [70], antibacterial properties [71], mechanical stability [64], and thermal properties [72], as well as material tunability, make them ideal for applications in the development of high-performance membrane technologies for liquid handling. These include ion removal [73], nanofiltration of organic solvents [74], reverse osmosis [75], and distillation in solar thermal systems [76]. However, only one type of MXene, Ti_3_C_2_T_x_, has been used for membrane separation, highlighting the early stage of MXene research in water purification. To advance research in this area, it is suggested that the specific structural type of MXene used in the work should be reported rather than the family name (MXene).

### 2.1. Common MXene Synthesis Procedure

The synthesis process of MXenes typically involves wet etching from the precursor MAX phase, followed by the layering of bulk crystals into monolayer or multilayer MXenes (Figure 3). Hydrofluoric acid (HF) is often used for this purpose as an etchant. This MXene synthesis process consists of three stages: etching, intercalation, and delamination. In the first step, the A layers in the MAX phases react with an acid, conventionally HF, to form MXene layers. In the second step, the A atoms in the MAX phases are replaced by hydrophilic groups -OH, =O, and/or -F, giving the MXene surface a negative charge and favoring the formation of stable dispersions. However, in the final stage, the bonding strength between the M_n+1_X_n_ layers is weakened by ultrasonic treatment and MXenes with loosely packed layered structures are formed [77].

Ti_3_AlC_2_ is the MAX precursor most commonly used to obtain the most studied Ti_3_C_2_T_x_ MXene. This HF etching method has also been used to synthesize many new MXenes. However, the use of HF etchant carries certain hazards and requires special care when handling. In addition, this method is not effective for the preparation of stable nitride-based MXenes (Ti_n+1_N_n_). Therefore, the development of more versatile and environmentally friendly methods for the synthesis of MXenes is necessary.

### 2.2. In-Situ Synthesis of MXene

As the use of concentrated HF is associated with hazards, an alternative synthesis method has been proposed using in situ etching. In this process, the etchant is produced by the interaction of a fluoride salt and an acid. For example, with LiF and HCl, the inherent Li^+^ intercalation increases the distance between the MXene layers, facilitating delamination of the MXene sheets [80]. The same technology uses calcium fluoride (CaF), potassium iodide (KI), cobalt fluoride (NaF), lithium fluoride (LiF), and cesium fluoride (CsF) with hydrochloric acid (HCl) or sulphuric acid (H_2_SO_4_) to prepare the etchants [81,82,83].

This eliminates the need for an intercalation step. In addition, this process results in MXene sheets with larger lateral dimensions, as only a relatively mild ultrasonic treatment is required for delamination compared to HF etching. Furthermore, these technologies allow the production of multilayer transition metal carbides with different compositions and modified physicochemical properties. For example, the use of a mixture of LiF and HCl allows the production of higher quality monolayer Ti_3_C_2_T_x_ MXenes with defect-free surfaces compared to MXenes obtained by soft-etching (Figure 3f) [79]. Similarly, the use of milder etching solutions such as ammonium bifluoride (NH_4_HF_2_) facilitates the formation of Ti_3_C_2_T_x_ polymers with -NH_4_^+^ interlayers, which can be effectively used to remove negatively charged contaminants from water [84].

Despite these advantages, the in-situ method, being gentler than HF-based treatments, may leave unreacted MAX phases in the product, which need to be removed by centrifugation. Additionally, synthesis temperature and product washing significantly affect the MXene structure, as demonstrated by the study authors (Figure 3d,e) [79]. The exfoliation coefficient increases at lower etching temperatures, as does washing with ethanol, which has larger molecules than distilled water. Therefore, when using this synthesis method, it is important to consider the influence of various factors based on the intended application of the MXene.

### 2.3. Hydrothermal Synthesis of MXene

Hydrothermal processes are also used for the synthesis of MXene, with the notable feature that the use of extremely toxic vapors is excluded. For example, etchants such as NH_4_F can be used to hydrothermally treat Ti_3_AlC_2_ sheets to form Ti_3_C_2_T_x_. Studies have shown that Ti_3_C_2_ and Nb_2_C MXenes can be synthesized by hydrothermal etching with NaBF_4_ and HCl without the use of HF [85]. Furthermore, hydrothermally synthesized MXenes have several advantages over those obtained by HF etching, such as higher group removal efficiency, larger interlayer spacing, easier delamination, and higher c-lattice parameter [86]. While the choice of salt has little effect on the MXene structures, Guo Y. et al. [87] used LiF, NaF, KF, and NH_4_F with HCl for the synthesis of Mo_2_CT_x_ MXene. The resulting MXene morphologies were similar, with an Mo content variation of 3.6–7%. However, the choice of salt influences the optimal synthesis temperature depending on its solubility and the radius of the hydrated cations.

A similar scheme applies to other synthesis methods, such as microwave assisted synthesis, where microwave radiation is used instead of ultrasound after etching. This suggests that there are many ways to weaken the M-X bond, and the effects of these methods on the MXene structure should be considered [88].

### 2.4. Fluorine-Free Synthesis of MXene

Recent MXene synthesis methods such as molten salt [89], acoustic synthesis [90], and chemical vapor deposition (CVD) [91] do not require the use of hydrofluoric acid. These methods represent the latest advances in MXene synthesis. The A layers of the MAX phase can also be leached out following hydrothermal treatment in H_2_SO_4_, as demonstrated by Cai Y. et al. [92] for the synthesis of Ti_3_C_2_. However, this method is time consuming. Therefore, the hydrothermal method needs to be optimized by increasing the solution concentrations to reduce the synthesis time and allow scaling up.

There are also alternative methods, such as anodic corrosion of Ti_3_AlC_2_, where a salt solution containing chloride and hydroxide ions is prepared. In this scheme, cations and the resulting neutral compounds are absorbed by the MAX phase, increasing the interlayer distance and allowing chloride ions to penetrate and etch the A layer. The use of other anions remains questionable, as complete etching of the Al layer from Ti_3_AlC_2_ using anions other than chloride ions has not been successfully achieved [93]. Therefore, despite the prospects and potential of these technologies, they have yet to be effectively demonstrated for large-scale application.

## 3. Properties and Synthesis of MXene/Carbon Nanocomposite

The use of MXene nanosheets in the active selective layer can significantly improve membrane performance for forward osmosis. For example, in the study [94], the optimum aqueous flux of the membrane modified with MXene nanosheets increased by approximately 80%, while the reverse solute flux remained minimal. This demonstrates that the incorporation of MXene nanosheets can enhance membrane performance. However, significant challenges remain regarding the flexibility, functionality, and stability of MXenes. MXene nanosheets inevitably undergo irreversible stacking due to their high surface energy and interfacial van der Waals interactions. Additionally, the weak gel-forming ability and high mechanical stiffness of MXene sheets create significant difficulties in assembling macroscopic monolithic structures based on them. Therefore, improvements are needed, including the application of hybridization methods, appropriate functionalization/modification, and optimization of synthesis/reaction conditions. In addition, low water flux is one of the main problems preventing the use of MXene membranes in large scale water treatment systems. However, the water flux through MXene membranes can be significantly increased by increasing the layer spacing [95]. In this context, numerous efforts have been made to create three-dimensional macroarchitectures of MXene-based materials, and various types of MXene-based composites have been developed, such as MXene/polymer [45,96,97,98,99] and MXene/metal oxide composites [44,100,101,102,103]. Polymers are often used as binders in the design of three-dimensional macroscopic materials [104], but this inevitably leads to resistance to interfacial contact.

To address these issues while considering the importance of environmental safety and unique structure, carbon-based nanomaterials and their application in composites with MXenes are being actively investigated for further use in water purification from various contaminants.

### 3.1. MXene/Carbon Nanotube Composite

Carbon nanotubes (CNTs) with a suitable surface area [105], multifunctionality, and needle-like shape [106] can be produced by carbon arc discharge [107], spray pyrolysis [108], chemical vapor deposition (CVD) [109], plasma synthesis [110], laser ablation [111], and other methods. CNTs possess several unique properties, including good thermal conductivity (558.06–700.15 W/m·K [112]), high tensile strength (>150 GPa, Young’s modulus = 1 TPa [113]), flexibility, hollow monolithic structures [114], suitable permeability [115], electrochemical properties [116], and ballistic transport properties [117], making them promising candidates for environmental protection applications.

In view of these properties, recently developed MXene/CNT hybrid composites exhibit remarkable electrochemical and adsorption properties as well as unique mechanical properties [118,119,120,121,122,123,124,125,126,127,128,129,130,131,132,133]. These composites have new possibilities for creating materials with a variety of applications, including sensing, catalysis, water desalination, pollutant removal, electromagnetic interference shielding, and applications in supercapacitors and batteries.

Hybridization of MXene with CNTs can improve their mechanical properties and resistance to aggressive environments such as alkali solutions. For example, Qian Y. et al. [118] developed three-dimensional (3D) nanofillers of MXene (Ti_3_C_2_T_x_)/CNT with high stability and improved interfacial adhesion between adjacent fiber filaments. They also exhibit increased tensile and flexural strength, with excellent surface roughness and flexural strength even after 120 days of immersion in alkaline solutions.

In MXene-based membranes, improved interfacial adhesion is achieved through hybridization with CNTs. This is facilitated by stable π-π interactions and van der Waals forces of CNTs, which promote tight packing of MXene nanosheets. For example, one study [119] indicates that the opposite charge between the MXene solution and the CNT-cetyltrimethylammonium bromide solution enhances the electrostatic attraction between adjacent nanosheets, which also improves the adhesion strength at the phase interface and increases the efficiency of the MXene/CNT-cetyltrimethylammonium bromide membrane. The researchers suggest that the slits formed by a large number of intertwined CNTs with hydrophobic inner and outer surfaces, which act as frictionless flow channels that facilitate the acceleration of water sliding, explain the mechanism of rapid water transfer. The CNT-cetyltrimethylammonium bromide imparts a columnar structure to the membrane, which has a larger d-interval for rapid molecular transport.

The properties of MXene/CNT composites depend significantly on the synthesis method, which influences the composite structure and/or CNT growth. The synthesis scheme and the influence of the synthesis method on the composite morphology are shown in Figure 4 and Figure 5, respectively.

Among the various methods, the self-assembly method stands out for its accessibility and practicality. This method relies on the electrostatic interactions between oppositely charged MXenes and CNTs, which simplifies the process by eliminating the need for complex equipment or conditions. The self-assembly process can be easily implemented at room temperature, making it energy-efficient and cost-effective. Moreover, this method allows for precise control over the composite structure by adjusting the surface charges and interaction parameters. Modification of CNT surfaces with agents like polyethyleneimine enhances the electrostatic interactions, further facilitating the uniform and stable integration of MXenes and CNTs. This simplicity and effectiveness in creating well-defined composite structures make the self-assembly method particularly appealing for practical applications. Furthermore, self-assembling composite membranes demonstrate excellent selectivity, high operational efficiency, and environmental sustainability, which explains their polarity [120,121,134].

However, the presence of chemical bonds between the components can provide greater stability to the composite and prevent degradation in aqueous environments. This has led to the exploration of other MXene/CNT synthesis methods. Microwave irradiation can be used to grow CNTs on MXene in the presence of a catalyst. MXene substrates are critical for CNT growth as they provide nucleation sites due to their unique layered structure, high thermal conductivity, and large surface area.

The grown CNTs and metal oxides serve as deposition sites that promote further growth of the CNTs or carbon nanoparticles. This represents a new approach to the synthesis of hybrid nanostructures based on CNTs and MXene. For example, Gao X. et al. [113] showed that the catalyst can be removed by subsequent crosslinking of the CNTs. The CVD method, using organic gases such as acetylene as a carbon source, can provide a similar CNT growth scheme on active sites, where the active site is a pre-existing early transition metal [123].

Moreover, chemical crosslinking of materials with MXene during membrane construction increases water treatment efficiency and can improve performance and fouling resistance, as demonstrated in the case of MXene (Ti_3_C_2_T_x_)/cellulose acetate mixed matrix membrane [135]. Thus, the membrane synthesis method plays a key role in realizing the potential of the membrane and its composites.

The significant problem of material swelling that occurs when using MXene and its composites for water purification remains a major challenge. This swelling impedes the flow of water through the material and limits the potential for effective reuse. For example, in [136], MXene nanosheets were added to a thin-film composite membrane to reduce mass transfer resistance and improve membrane performance. Despite promising results showing that the water contact angle decreased by about 16%, indicating an increase in hydrophilicity, the swelling of MXene nanosheets and their agglomeration remained problems, limiting further improvement in membrane performance; the MXene/CNT composite presented by Zheng W. et al. [124] has a unique structure including bridges, which exhibits stability and resists swelling when in contact with water.

Researchers are also actively exploring the use of matrices, substrates, and membranes in which the composite is held during deposition or formation. This approach allows for the control of the composite’s flexibility, interlayer spacing, and structural features. This principle can be applied to the development of materials for water purification using MXene/CNT composites. Examples of such work are shown in Table 1.

There are numerous synthesis methods for MXene/CNT composites, including mechanical mixing, self-assembly, co-dispersion, electrophoretic deposition, CNT growth on active sites via CVD, thermal treatment, microwave processing, and hydrothermal processes. Among the various MXene/CNT-based architectures, aerogels and foams with three-dimensional structures are particularly attractive candidates due to their unique mechanical properties and excellent permeability for gases and liquids. This opens up broad prospects for various applications, including water purification.

Synergistic interactions between MXene and CNTs in hybrid structures can mitigate or prevent serious problems associated with the restacking of two-dimensional structures, thereby enhancing their properties for various applications. The three-dimensional structure can be achieved by adding other substances besides CNTs, such as metal oxides, as demonstrated by Zheng W. et al. [124]. Additionally, Zhang Z. et al. [134] used MXene modification with alkali to increase the surface functional groups, which improved its hydrophilicity. Sodium lignosulfonate-CNTs serve as a hydrophilic bridge, effectively connecting the substrate membrane to the MXene layer. Consequently, the interaction of these components resulted in improvements in membrane properties such as adsorption capacity, stability, chemical resistance, and corrosion resistance. Their synergistic effect increased the rejection of pollutant molecules and water permeability. In this regard, it is important to consider which additives will have a beneficial effect in obtaining a high-performance composite suitable for water purification and the availability of materials for the widespread application of the composite.

Currently, the study of MXene/CNT composites for their application in electronic devices is much more popular than for water purification, despite the morphological features and potential of the material. This area requires further investigation to obtain more accurate data on the adsorption characteristics of the material.

In addition, several other challenges could hinder the practical application of MXene/CNT composites for water purification. First, ready-made CNTs typically contain metallic impurities (Fe, Co, etc.) derived from catalysts used during synthesis [131,132]. Although acid treatment is effective in removing these metallic nanoparticles, this process can lead to defects and a reduction in the aspect ratio of the CNTs. Second, single-walled CNTs exist in a wide variety of structures and diameters, and the sorting process inevitably results in high-purity CNTs, whereas selective growth does not guarantee the same level of purity. Third, the inert nature of defect-free CNTs makes it difficult to form strong chemical bonds with other materials, complicating the achievement of a stable dispersion for liquid processing.

Therefore, individual nanomaterials (whether MXene or CNTs) may not be sufficient for the production of high-quality membranes for water purification. This necessitates the development of a rational design for MXene/CNT hybrids embedded in a specific matrix that maximizes the synergistic effect of each component in the system. To overcome the serious problems associated with restacking and swelling of MXene/CNT-based composites, which occur during the manufacture of two-dimensional (2D) sheet membranes and separation processes, new methods are required to improve the performance and quality of the product. It is essential that the composite retains its adsorption capacity and sensitivity, so these issues need to be addressed comprehensively, taking into account various aspects of its functionality and application.

### 3.2. MXene/Carbon Nanofibers Composite

Carbon nanofibers (CNFs) have already been successfully used for water purification [133]. Fan Q. et al. [137] developed a complex composite, Co_3_O_4_@CNF/persulfate (2KHSO_5_·KHSO_4_·K_2_SO_4_), for the removal of organic pollutants from wastewater, achieving removal rates of 46.5% and 58.4%. In this field, CNFs are used due to their special properties such as high adsorption capacity (827.5 mgNaCl/gCNF [138]), chemical stability [139], and large specific surface area (973.4 m^2^/g [140]). In addition, they exhibit catalytic activity as catalyst supports [141]. Interestingly, the application of MXene with CNFs has been noted to form flexible, durable, and porous composites that are widely used in the electronics industry [142]. In the MXene/CNF composite, various forces such as Van der Waals forces, hydrogen bonds, ionic bonds, and covalent bonds play a role in influencing the interfacial adhesion and mechanical properties of the composites. Hu S. et al. [143] found that covalent bonds provide carbon fiber (CF)-MXene with maximum surface roughness, active functional groups, and surface energy, which positively affect the adsorption properties of the composite.

Electrophoretic deposition can be used to create a composite, as in [144] (the scheme is shown in Figure 6). This method was used to improve the interfacial properties by uniformly coating the surface of CFs with two-dimensional MXene (Ti_3_C_2_T_x_) nanoparticles. Studies have shown that the presence of MXene nanoparticles on the surface of carbon fibers significantly increases their surface energy, wettability, and roughness, leading to a noticeable increase in the interfacial strength and flexural properties of the carbon composite. This versatile approach, based on the combination of electrophoresis and epoxy resin, promises to be effective in modifying carbon composites, potentially leading to a significant improvement in performance. It is noteworthy that due to the high surface energy of MXene in hydrophilic membranes, the membrane primarily interacts with water molecules during filtration, forming a hydration layer on its surface. Contaminants must penetrate or disrupt this hydration layer before they can contact the membrane surface, thereby enhancing the membrane’s anti-fouling properties [69].

The self-assembly method was used to synthesize CF/MXene composites [145], as was the case for MXene/CNT composites [120]. Zhao X. et al. [145] used amino-functionalized CFs coated with Ti_3_C_2_T_x_ MXene nanosheets by electrostatic assembly and subsequently coated with epoxy resin. This work is unique in its structural characteristics of MXene/CF (Figure 7f). The results show that Ti_3_C_2_T_x_ nanosheets are coated and firmly attached to the surfaces of NH_2_–CFs, forming a three-dimensional network structure. Ti_3_C_2_T_x_ nanosheets demonstrate the potential to significantly improve the mechanical properties of CF composites when used as a reinforcing element firmly attached to the CFs. In the resulting composite, an increase in interfacial adhesion at the NH_2_–CF/MXene/epoxy resin interface was observed, together with a 40.8% increase in tensile strength, a 45.9% increase in compressive strength, a 38.5% increase in shear strength, and a 74.4% improvement in impact toughness compared to untreated CF/epoxy resin composites. The Ti_3_C_2_T_x_ nanosheets are entirely responsible for the improvement in mechanical properties, acting as an additional reinforcing layer at the phase boundary.

Another relatively simple method of obtaining the composite is thermal treatment. Sun L. et al. [146] carried out such a synthesis. The analysis results of the hierarchical porous MXene/CF heterostructure, similar to a “shell/skeleton”, obtained by one-step pyrolysis of a commercial windscreen cleaner made of cotton fibers impregnated with Ti_3_C_2_T_x_ solution show that this composite combines the stable and flexible properties of CF with the excellent electrical conductivity of Ti_3_C_2_T_x_. MXene nanosheets embedded in the porous CF mesh successfully prevent self-stacking and aggregation, ensuring rapid diffusion and electrolyte ion transfer. The heterostructure also reduces internal resistance, which contributes to efficient and fast electron transfer during charge/discharge processes, resulting in high performance.

Hydrogels and aerogels based on MXene/CNF composites can be effectively used for water purification. Chao M. et al. [147] developed a Ti_3_C_2_T_x_/shortened CF/polyimide aerogel using lyophilization and thermal imidization methods. This product shows excellent mechanical properties, flame retardancy, and thermal stability. However, the surface should be modified from hydrophobic to hydrophilic properties for use in aqueous environments.

A widely used method for obtaining MXene/CNF composites in practice is vacuum filtration. The composite MXene/CNF/parous carbon film obtained by this method showed excellent flexibility and porosity (574.5 m^2^/g) [148]. Also, the self-cleaning effect of MXene was observed in a composite with titanium oxide nanotubes, as reported in [149]. This suggests another promising direction for developing a membrane based on MXene and CNFs with significant benefits. This improvement is due to the incorporation of CNFs into the composite, which act as nanoscale spacers, forming an “interconnected topology” between MXene layers and significantly increasing the mechanical strength of MXene-based membranes. This underscores the advantages of using MXene and CNFs in composites for water purification. The addition of metal oxides can further enhance the water purification efficiency of MXene membranes. According to the authors, based on the extended Derjaguin–Landau–Verwey–Overbeek theory, the incorporation of TiO_2_ nanotubes increased the interaction energy between the MXene/TiO_2_ nanotubes membrane and contaminants, thereby improving anti-fouling properties [149].

CNF is also known to have binding and stabilisation properties. Qian K. et al. [150] successfully synthesised spherical microspheres of CNF and CNTs obtained by the spray method. These microspheres were mixed with MXene to form a composite by a pressure extrusion method and then underwent an annealing process. As a result, hybrid films with high conductivity were obtained, consisting of multilayer Ti_3_C_2_T_x_ in which filamentous yarn-ball-shaped microspheres of CNFs/multi-walled CNTs were embedded. CNFs were used for stability as a binder for CNTs and MXene. The composite exhibited enhanced absorption capacity due to the improved polarisation at the interface and the complex spherical structure.

**Figure 6 membranes-14-00184-f006:**
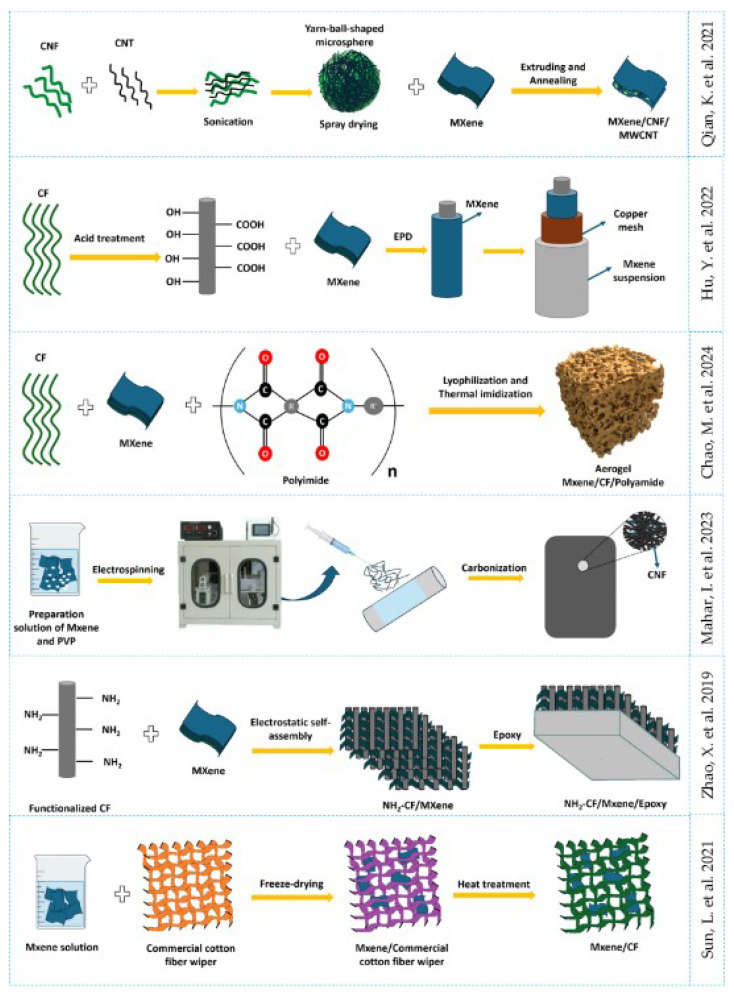
Methods for synthesis of the MXene/CNF composite (adapted with permission from references on the right) [144,145,146,147,150,151].

**Figure 7 membranes-14-00184-f007:**
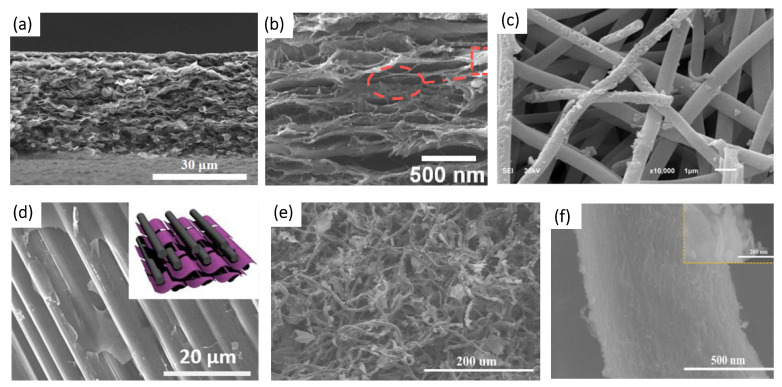
Morphologies of MXene/CNF composites obtained by different methods: (**a**) Ti_3_C_2_T_x_/CNF/multi-walled CNT [150]; (**b**) d-Ti_3_C_2_T_x_/CF membrane [144]; (**c**) Ti_3_C_2_T_x_/CNFs composite membrane [151]; (**d**) NH_2_-CF/Ti_3_C_2_T_x_ [145]; (**e**,**f**) Ti_3_C_2_T_x_/CF [146].

Nevertheless, research on the application of MXene/CNF composites for the removal of heavy metal ions from water has been sparse until recently. Yan S. et al. [152] reported the synthesis of FeCo_2_S_4_/MXene/porous CNF by electrospinning, using peroxyacetyl nitrate and polyvinylpyrrolidone as precursors for porous CNF. The product exhibited high electrochemical dynamic properties due to modification and structural design. Even without modification, the product showed excellent stability, demonstrating the potential of this method for preparing composites for water purification, as highlighted by Mahar I. et al. [151]. In this work, a membrane of MXene/carbon nanofiber (CNF) composite nanofibers was formed using polyvinylpyrrolidone and electrospinning, which represents an effective strategy for capturing toxic elements from aqueous solutions. The interaction between the functional groups of polyvinylpyrrolidone and MXene forms strong bonds that contribute to the formation of a stable membrane. This approach ensures high mechanical strength, a microporous structure, a large specific surface area, potential for multiple uses, and the stability of carbon nanofibers. The authors highlight the potential use of active sites on the MXene surface in combination with carbon nanofibers to improve selectivity towards heavy metals, minimize human health risks, and protect aquatic flora and fauna.

In view of this, MXene was synthesized from MAX phase Ti_3_AlC_2_ and incorporated into polyvinylpyrrolidone nanofibers, followed by carbonization to obtain a MXene/CNF composite for heavy metal adsorption. In this study, the adsorption efficiency of the MXene/CNF composite membrane was demonstrated for Pb^2+^ and As^3+^ ions at low concentrations (from 10 to 10 ppm). These impressive results indicate that nanofiber membranes based on MXene/CNF composites pave the way for an effective, affordable, lightweight, and environmentally friendly method of reducing Pb^2+^ and As^3+^ ions to acceptable levels for human consumption. The potential of this material also can be applied to water purification from other metals. Further research is needed to optimize the membrane production process for wider applications.

### 3.3. Composite MXene/Graphene Oxide

Graphene oxide (GO) is a two-dimensional nanomaterial consisting of graphene sheets covalently modified with oxygen atoms, which form various functional groups. Due to its high adsorption capacity, GO holds promise for water purification applications, including membranes [153]. Its magnetic properties, induced by the addition of metal oxides [154,155,156,157], are also gaining attention in modern research. Hybrid layered structures composed of stable monolayer dispersions of graphene oxide and MXene, with tailored functional groups, offer a promising approach for enhancing water purification efficiency. The crystalline structure of MXene differs significantly from that of other two-dimensional materials, including graphene oxide, which consists of amorphous oxidized sp³ carbon domains and hexagonal sp^2^ carbon domains [158]. GO has a relatively large, irregular interlayer spacing that requires additional processing, similar to its disorientated surface structure.

Various types of GO surface modifications have been developed for additional surface tuning, including hydroxyl-dominant GO, epoxy-dominant GO, carboxyl-dominant GO, reduced GO, and others [153,159]. As a result, GO exhibits instability and a tendency to degrade in water due to electrostatic repulsion between GO nanoparticles. In contrast, MXene has an ordered surface with some point defects and broken bonds, which allows for the tuning of narrow interlayer spacings suitable for membrane applications. Therefore, MXene and GO can complement each other in composites, potentially leading to more efficient water purification. Notably, GO is unique as a functional two-dimensional material with tunable optical, thermal, and hydrophilic properties due to its simple modifiable surface chemistry [153,156]. This opens up possibilities for creating complex multi-component materials. Additionally, MXene’s restorative capability can serve as a basis for rational composite assembly and regulation, allowing control over the formation of three-dimensional functional structures.

Ming X. et al. [160] developed macroscopic porous GO/Ti_3_C_2_T_x_ aerogels with hierarchical nanostructures using directional sublimation drying and post-ion crosslinking (Figure 8). The formation of the aerogels involves ice crystals, GO sheets, and long polymer chains of sodium alginate interacting through π-π interactions and hydrogen bonding of functional groups. These elements act as binding ‘bridges’ that facilitate the formation of three-dimensional structures from MXene sheets, thereby improving the thermal conductivity and mechanical properties of the aerogel. Ion crosslinking occurs by interaction of gelatin and water-insoluble calcium alginate and imparts mechanical strength to the resulting GO/Ti_3_C_2_T_x_ aerogel. This approach, which differs from traditional covalent bonding methods, combines the simplicity of ion crosslinking with efficient porous structure formation by freezing, ensuring rapid and reproducible synthesis. The resulting GO/Ti_3_C_2_T_x_ aerogel has a multi-level porous structure and ultra-light properties, making it promising for various technological and industrial applications, including water purification and solar energy conversion. Meanwhile, Liu T. et al. [161] demonstrated that GO/MXene composite membranes synthesized by filtration exhibit high efficiency in improving water flux and removing organic pollutants such as dyes and natural organic matter present in raw water. Comparing membranes based on GO or MXene with heterogeneous composite structures of GO/MXene, the authors note that the combination of the two components in the composite provides a synergistic effect, optimizing the balance between selectivity and permeability. The water flux through the GO/MXene composite membrane reaches 71.9 L/m^2^·h·bar, significantly exceeding the flux through the pure GO membrane, which is 6.5 L/m^2^·h·bar. This effect is explained by the formation of low friction nanocapillary channels due to the addition of MXene and the combination of molecular sieving effects and electrostatic interaction of the composite materials.

Composite GO/MXene membranes exhibit high permeability due to increased interlayer spacing and reduced content of oxygen-containing functional groups. This promotes the formation of multi-stage micropores, prevents the accumulation of plate-like nanosheets, and facilitates water transport through microchannels. Controlling the content of surface functional groups in such membranes helps maintain the high selectivity characteristic of GO membranes and reduces unwanted interactions between the membrane surface and water molecules, thereby decreasing water flow resistance. This highlights the importance of the presence of GO in the composite. The GO/MXene composite membranes developed in this study thus overcame the issues of low flux and instability typically associated with pure GO membranes.

The literature highlights that membranes based on GO and MXene can exhibit hydrophilic properties depending on the contact angle with water, probably due to the presence of different functional groups such as -COOH, -OH, and epoxy groups on the surface of graphene oxide nanosheets [153,154,155,156,157,158]. At the same time, the presence of relatively hydrophobic MXene may reduce the overall hydrophilicity of the membrane. One of the main characteristics of the Ti_3_C_2_T_x_/GO membrane is the formation of non-selective channels, which can be explained by the small lateral size of the exfoliated MXene nanosheets or residual multilayer sheets after exfoliation, which hinders the ordered arrangement of two-dimensional materials.

Other components may also be added, depending on the nature of the contamination. For example, a graphene oxide membrane intercalated with TiO_2_ nanocrystals was prepared by oxidizing Ti_3_C_2_T_x_ nanoparticles with H_2_O_2_ in situ, followed by vacuum filtration on a membrane made of a complex cellulose ether mixture for the ultrasensitive detection of uric acid [162]. The GO-Ti_3_C_2_T_x_-H_2_O_2_ membrane showed superior permeability and efficiency in removing contaminants compared to the original graphene oxide membrane and the GO-TiO_2_ membrane.

When fabricating membranes from GO and MXene, it is important to consider that the presence of a large number of oxygen-containing functional groups on GO nanosheets makes the membranes highly hydrophilic and prone to swelling in humid or aqueous environments, which can lead to poor stability. When immersed in an aqueous solution, oxygen-containing functional groups such as hydroxyl and carboxyl groups on GO nanosheets deprotonate, causing electrostatic repulsion between adjacent GO nanosheets that exceeds the π-π interactions between them. Therefore, selecting the optimal GO content in the membrane is crucial.

Composite GO/MXene membranes with optimal mass ratios remain stable under aqueous conditions, which is attributed to the reduced electrostatic repulsive forces due to significantly fewer surface functional groups in MXene. The water permeability of the membrane also increases with increasing MXene content for several reasons. First, the π-π interactions between successive GO nanosheets are reduced due to the presence of MXene, which increases the interlayer distances and allows greater water flow. Second, the interaction between water molecules and oxygen-containing functional groups on the oxidized regions of GO nanosheets is weakened by the inclusion of MXene, reducing water flow resistance and increasing permeability. Thus, membranes based on GO and MXene possess several advantageous properties that make them promising for the efficient purification of water from heavy metals and salts. These composites represent valuable materials capable of addressing future challenges associated with freshwater scarcity.

### 3.4. Composite MXene/Organic Based Material

MXene-based aerogels are also notable for their functionality, but technical issues with their filling and stabilization complicate their application [163,164,165]. Cellulose nanofibers (CeNFs) are currently being explored for the scalable production of durable aerogels with efficient adsorption capacities. MXene/CeNF nanocomposites represent a new type of material with unique properties that can be utilized in various applications, from electronics to sensors. They also offer potential for unconventional uses as a water filtration material, showing numerous advantages over traditional graphene and activated carbon filters due to the unique combination of layered MXene structures and porous nanocellulose [166]. They are also noted for their exceptional adsorption capacity, as demonstrated by Akhlamadi G. et al. [163]. Ti_3_C_2_T_x_/CeNFs aerogel has ultra-high sorption capacity towards various oils/organic solvents, ranging from 110 to 320 times its weight. This allows a wide range of contaminants to be effectively removed from water, including heavy metals, organic compounds, and micro-organisms. Such composites possess high mechanical strength, flexibility, and remarkable lightness, making them resistant to damage and wear [166]. They can be formed into various shapes and configurations, which enhances their practical applications in water treatment systems. MXene’s diverse compositions, which affect its properties, make it versatile for addressing different types of pollutants. For example, its antibacterial properties can prevent the growth of bacteria and other microorganisms on the filter materials [71], reducing the need for frequent filter changes and improving the quality of the treated water. Nanocellulose-based nanocomposites are environmentally friendly and biocompatible, ensuring safety for the environment. Additionally, they exhibit high chemical and thermal stability, which contributes to their durability under operational conditions [167]. This reduces the need for maintenance and replacement of filters, making them more cost-effective in the long term.

Wu N. et al. [168] developed the fabrication of MXene-based aerogels dried at ambient pressure (Figure 9). These aerogels are characterized by a large surface area, ultra-light weight, and high strength, thanks to the use of ultra-thin, high-strength, and renewable cellulose nanofibers (CeNFs) as highly efficient crosslinking materials for MXene nanosheets.

Two unique approaches in the creation of these aerogels are noted: the application of an ambient pressure drying method and the use of CeNFs. The atmospheric pressure drying method enables environmentally friendly, simple, and scalable production over large areas without the need for energy-intensive equipment, special gases, or high temperatures. Additionally, the multiple functional groups on CeNFs are effective for chemically crosslinking MXene-based aerogels, significantly improving their oxidation resistance. The ultra-thin 1D CeNFs (with an average diameter of only 1.4 nm) provide minimal gaps between the highly conductive MXene nanosheets, maintaining the high electrical conductivity of the aerogels, while the numerous heterogeneous interfaces ensure excellent conductivity.

The strong interaction between MXene and CeNF promoted the formation of mechanically strong cell walls, which ensured the stability and strength of the aerogel structure during the drying process. Consequently, MXene aerogels created using the ambient pressure drying method, with high porosity, exhibited multifunctionality, including efficient dye adsorption, remarkable photothermal oil absorption, and electromagnetic interference protection [168]. Additionally, the aerogel density and MXene content can be adjusted over a wide range, allowing for diverse applications and material characterization. This work opens up new possibilities for the preparation of environmentally friendly, reliable, and multifunctional MXene-based aerogels via ambient pressure drying, with significant potential for water purification applications.

Ti_3_C_2_T_x_-based biomimetic hybrid aerogels with oriented tracheid-like textures can also be prepared using a simple bidirectional freezing technique with the addition of CeNF and CNTs according to Xu T. et al. [169] (Figure 10). Curiously, the interwoven “mortars” of CeNF and CNTs bonded to MXene “bricks” of tracheid structure by synergistic electrostatic interaction and hydrogen bonding provide excellent interfacial bonding and superior mechanical strength. These aerogels exhibit compressibility of up to 80% and exceptional durability, withstanding 1000 cycles at 50% strain.

Wu F. et al. [170] employed a method combining electrospinning, freezing, and subsequent carbonization to produce MXene/C aerogels using polyacrylonitrile nanofibers. The resulting microwave-absorbing aerogel exhibited a three-dimensional network structure characterized by high porosity and low density. These structural properties underscore the potential applications of carbon and activated carbon in water purification. The exceptional attributes of this aerogel highlight its significant potential for various applications, particularly in the field of water purification.

Thus, aerogels based on CeNF and MXene are innovative materials with great potential for water treatment. However, further research is needed to evaluate their effectiveness against various types of pollutants and to explore integrated removal strategies. In addition, practical applications require addressing cost, energy consumption, and scalability issues, which remain uncertain. Therefore, new approaches are needed to overcome the limitations of MXene/CeNF aerogels.

It is also worth noting that MXene synthesis for water purification purposes typically employs one of three methods: HF etching [79], in-situ synthesis [171], and, in rare cases, hydrothermal synthesis [85,86] or fluorine-free synthesis methods [89]. Additionally, composites with carbon nanomaterials primarily use MXenes synthesized via HF etching [122,123,127] and in-situ methods [125,161,162]. This is because these methods allow for more thorough delamination of A layers from the MAX phase, which subsequently affects the quality of the composites. Therefore, the development of high-quality MXenes through safer methods than HF etching remains relevant, with the in-situ synthesis method being the only viable alternative at present.

The choice of methods for producing MXene/carbon nanocomposites also plays a crucial role in their water purification efficiency. Incorporating carbon nanomaterials and MXene into a composite can significantly enhance their properties, resulting in a synergistic effect where the combined performance exceeds the sum of each component’s individual properties. To achieve this effect and ensure that the components complement each other, it is essential to use appropriate composite fabrication methods. The specific choice of carbon nanomaterial and the intended application of the final material also influence the optimal fabrication approach. As discussed, membranes are ideally suited for CNTs, whereas aerogels are more suitable for CeNFs.

## 4. Mechanism of Water Purification from Various Pollutants

### 4.1. Metals

In the case of water purification from metals using MXene/carbon nanocomposites, the adsorption mechanism primarily involves the chemical interaction between functional groups and metal ions. This mechanism is considered the main sorption process, influenced by the number and nature of the functional groups involved. The higher efficiency observed can be attributed to the abundance of functional groups on MXenes as well as the layered and porous structure of the nanofibers. Therefore, modification of MXenes is promising for enhancing this process [172,173,174]. Silicon oxide and amino group functionalization are commonly used for such modifications [175]. Wan K. et al. [176] demonstrated that MXene and CeNF-based membranes effectively purify water from antimony ions, with β-FeOOH modification further enhancing the purification efficiency (Table 2). The mechanism of electrostatic adsorption of antimony ions by Ti_3_C_2_T_x_/CeNF/FeOOH membrane is shown in Figure 11. The mechanism of antimony (Sb) adsorption in this membrane primarily involves interactions between Sb species and the membrane components. β-FeOOH plays a crucial role as an active center, significantly enhancing Sb removal. A substantial portion of Sb is adsorbed onto β-FeOOH, with additional adsorption occurring on MXene. The interaction energy between β-FeOOH and Sb species is predominant, indicating strong binding and inhibition of Sb diffusion. The proposed mechanism includes surface coordination and electrostatic attraction of Fe-OH groups within the membrane, which effectively traps Sb species. MXene and CeNFs contribute to the membrane’s mechanical strength and stability across different pH levels. Additionally, the membrane’s selectivity and self-cleaning ability are noteworthy. Surface roughness, multilayer structure, and porosity are essential for enhancing the material’s adsorption capacity.

Xia Y. et al. [177] show that the Ti_3_C_2_T_x_-nanoribbons/CNT material is highly sensitive and selective for Hg^2+^, indicating the possibility of using the material for water purification from mercury ions. The adsorption and reduction reactions on the Ti_3_C_2_T_x_-nanoribbons cause Hg^2+^ to be reduced on the surface as Hg^0^. The authors point out that the same scheme can be used for the adsorption of Cd^2+^, Cu^2+^, Zn^2+^, K^+^, Mn^2+^, Ca^2+^, Mg^2+^, Co^2+^, and Fe^3+^.

The reducing properties of MXene have been demonstrated by Naji M.A et al. [178]. This study highlighted the high rejection rates of Cu^2^⁺, Cd^2^⁺, and Pb^2^⁺ ions, showcasing the effectiveness of MXene-based membranes. In this work, MXene membrane was modified with polyphenyl sulfone, resulting in a water flux ranging from 2.74 to 11.1 L/m^2^·h·bar.

On the other hand, in the membrane incorporating CNTs, the CNTs serve to isolate and support the MXene layers, which act as reducing agents. Polydopamine, a polymer used in this membrane, enhances the adhesion of the substrate to the Ti_3_C_2_T_x_ and CNT sheets [179]. This hybrid membrane exhibits exceptional performance, with a water permeability of 437.6 L/m^2^·h·bar (2.46 × 10⁻^18^ m^2^), which is approximately 202 times higher than that of a pure Ti_3_C_2_T_x_ membrane. Additionally, it shows excellent ability to trap and retain Au^3^⁺ ions, reducing them to Au^0^. This makes the membrane highly suitable for water purification. Moreover, the researchers successfully separated the precious metals deposited on the membrane’s surface, forming an Au film with some content of Ti_3_C_2_T_x_ and CNTs, using an electrochemical barbotage transfer method.

The mechanism of metal removal from water in this case is more complex than simple ion exchange processes. Theoretical studies indicate that the degree of oxidation of the terminal Ti atoms in Ti_3_C_2_T_x_ MXenes is unsaturated, allowing spontaneous electron transfer from these atoms to metal cations, thereby reducing them to metal nanoparticles. Research has demonstrated that MXenes with suitable terminations can effectively reduce noble metal ions to their zero-valent forms. While CNTs with functional groups can also facilitate metal reduction, their contribution is significantly less compared to Ti_3_C_2_T_x_ [179,180].

The spectra of the Ti_3_C_2_T_x_ membrane can be divided into four main components corresponding to the Ti-C, C-Ti-OH, C-Ti-O, and Ti-O-O peaks. However, after using the membrane and filtering the solution with metal ions, it is observed that the C-Ti-OH peak disappears while the fractions of the C-Ti-O and Ti-O-O components increase significantly. It is thus established that the surface Ti atom in Ti_3_C_2_T_x_ ending with the -OH functional group has a higher reducing capacity than the Ti_3_C_2_ atom ending with the -O functional group. The latter leaves a pristine Ti_3_C_2_T_x_, i.e., a polymer with electronically unsaturated terminal Ti atoms. While -OH can donate electrons to noble metal cations such as Au^3+^, the noble metal cations oxidize C-Ti-OH to C-Ti-O and/or Ti-O [77,177,178,179,180]. Consequently, the membranes based on Ti_3_C_2_T_x_ and carbon nanomaterial form a redox couple that rationalizes the spontaneous electron transfer from the membrane to the metal ions, resulting in the reduction of the ion to a zero-valent metal.

Moreover, when MXene/carbon material membranes are used, metal nanoparticles tend to accumulate on the membrane surface rather than penetrating inside. This behavior is attributed to the ultra-high electrical conductivity of MXene and CNT networks, which facilitates rapid electron transfer to the membrane’s surface during the redox reaction. Carbon materials, especially CNTs arranged in a three-dimensional framework, contribute to this electron transfer, allowing water to pass through while supporting the MXene sheets. This arrangement prevents aggregation of polymer layers and enhances the strength of the hybrid membrane [175,177,178,179].

The metal ion recycling mechanism using Ti_3_C_2_T_x_ demonstrates excellent capability for capturing metals from solution even at low concentrations. However, it is important to note that the degree of metal rejection diminishes for metals with lower electrode potentials [179]. In other words, metals with higher electrode potentials exhibit greater resistance to oxidation, which affects their removal efficiency.

### 4.2. Salts

Complex composites can be utilized to create hybrid systems combining direct osmosis and nanofiltration for wastewater treatment or targeted component extraction. Sun P. et al. [181] developed a thin-film nanocomposite membrane featuring a Ti_3_C_2_T_x_ MXene interlayer intercalated with CNTs (Figure 12a). This design enhances interfacial retention during degassing and increases monomer sorption through the amine interlayer, resulting in a higher degree of crosslinking and roughness in the membrane. Compared to traditional thin-film composite membranes without a multilayer structure, this membrane exhibits four times higher water flux and reduced specific salt flux. Notably, it demonstrated exceptional efficiency in concentrating real urban wastewater and significantly improved ammonia nitrogen removal, surpassing typical limits of permeability and selectivity. This study paves the way for the rational design and development of high-efficiency reverse osmosis membranes for environmental applications.

Thermal bonding between surface functional groups of MXene nanosheets and CNT leads to an overall reduction in membrane thickness, which can be explained by reactions such as -OH + -OH = -O^−^ + H_2_O and -COOH + -OH = -COO^−^ + H_2_O [127]. Thermal treatment of the material helps to prevent an increase in interlayer spacing in aqueous environments, thereby improving the rejection of specific contaminants. This process also reduces membrane swelling, increases water permeability by up to three times, and enhances contaminant retention [128]. The presence or absence of continuous 3D labyrinthine short mass transfer channels is crucial for effective water desalination. These channels can be created by integrating well-dispersed 1D carbon nanotubes with ultra-thin 2D MXene nanosheets, which ensures efficient water purification. Additionally, the thermal bonding temperature influences water permeability by affecting the height of the nanochannels and the hydrodynamic resistance to mass transfer [182,183]. Thus, it is important to find a balance between the membrane’s retention properties and its water and salt permeability.

Furthermore, negatively charged MXene/CNT membranes were found to reject salts with high selectivity based on Z^+^/Z^−^ values (charge ratio of cation to anion) in the order R (Na_2_SO_4_) > R (MgSO_4_) > R (NaCl) > R (MgCl_2_). The role of MXene in salt rejection is that MXene nanosheets are layered structures that can be stacked with interlayer spacings fine-tuned to allow water molecules to pass through while blocking larger salt ions. Additionally, MXenes often possess a negative surface charge due to the presence of functional groups like -OH, -O, and -F. This negative charge repels negatively charged salt ions (e.g., Cl⁻), reducing their passage through the membrane [126,184]. These results align with the Donnan exclusion theory, which indicates that electrostatic repulsion plays a significant role in ion rejection [185]. Consequently, the membrane’s charge can impact both water treatment efficiency and membrane selectivity.

The significance of binding energy in membranes is exemplified by the GO/etched-MXene membrane study [186]. In this study, MXene nanosheets were etched with hydrogen peroxide, which increased membrane stability and the number of mass transfer channels. As a result, the GO/etched-MXene membranes maintained stable separation rates during cross-flow filtration. The increase in the number of nanotransport channels and the distance between layers enabled the creation of a membrane with high separation efficiency, with CR/NaCl selectivity of 22.6 and CR/Na_2_SO_4_ selectivity of 42.0. This underscores the importance of channel quantity in enhancing water purification performance.

In addition, nanochannels in MXene-based composite membranes enable water to pass through while rejecting salts with larger molecular masses and sizes. For instance, the MXene/GO membrane used for purifying water from Na_2_SO_4_ achieved a salt rejection rate of 60.6% [184]. Thus, having a high density of nanochannels with smaller diameters facilitates more effective separation of salt ions from water molecules.

Thus, the performance of nanofiltration membranes is influenced by the charge density of functional groups and effective pore size. Surface potential measurements can characterize the charge density, which is affected by pH and salt type. The transport mechanism of ions through membranes involves charge and steric hindrance, quantified by salt retention at various pressures. Different salts can show different retention behaviors due to variations in charge and hydration radius. Some membranes have higher rejection of sodium salts, while others show higher rejection of magnesium salts. For example, GO membranes have larger pore sizes (0.6 nm) than the hydrated ion radius, reducing steric effects and emphasizing charge interactions. The GO membrane’s higher tortuosity increases diffusion length and effective charge density. Diffusion lengths can be adjusted by changing GO deposition, while interlayer spacing can be modified by altering the O/C ratio or crosslinking. Increasing diffusion lengths or reducing interlayer spacing enhances ionic retention due to increased diffusion/steric hindrance [187].

The MXene membrane surface also plays an important role in salt molecule rejection and desalination. The study by Xue Q. and Zhang K. [188] shows that the more the surface is negatively charged and the smaller the effective pore size, the higher the rejection. In a 28-day continuous desalination test, the enhanced negative charge MXene membrane designed by the authors achieved Na_2_SO₄ rejection rates of up to 98.6%.

In addition, a study by Kapitanov A. A. and Ryzhkov I. I. [189] presents a mathematical model for nanofiltration of binary aqueous electrolytes using MXene/CNT and PANi-PSS/CNT electrically conductive membranes. This model, based on the Nernst–Planck equations for ion fluxes, takes into account steric exclusion and Donnan exclusion caused by chemical and electronic surface charges. The model has been successfully applied to describe repulsion data for salts and anionic dyes. The membrane surface charge is generated through ion adsorption and functional group dissociation and is regulated by the potential of the electrically conducting membrane surface. The study also considers the effects of concentration polarization and shows that the effective membrane charge is significantly lower than the measured one, which is explained by complexation and/or adsorption of counterions. The results of the study provide new insights into the mechanisms of ion rejection by electrically conducting membranes.

### 4.3. Organic Contaminants

Another major issue in water treatment is the presence of organic contaminants such as solvents, oils, petroleum, and by-products from oil processing [190]. Challenges associated with this include low water flux, low rejection rates, and susceptibility to membrane fouling. MXene-based membranes are a promising solution to these problems. The large surface area of MXene facilitates the effective adsorption of various contaminants, including organic compounds and microorganisms [191,192,193,194,195,196,197]. MXene can adsorb organic contaminants through mechanisms such as π-π interactions, hydrogen bonding, and van der Waals forces, enabling the removal of substances like pesticides, dyes, pharmaceutical residues, and other organic compounds [119,191]. In the context of using MXene/carbon nanomaterial membranes for water purification from organic pollutants, the interlayer distance is a critical factor influenced by the mass content of carbon-based materials and MXene. Yang L. et al. [194] demonstrated that by incorporating CNTs with Ti_3_C_2_T_x_ and applying synergistic modifications with polydopamine and polyethyleneimine, the membrane achieved superhydrophilicity, superoleophobicity, and high efficiency in separating oil-water emulsions (>99.9%). The membrane withstands high permeation fluxes (over 4432.13 L/m^2^ h·bar), is resistant to fouling, and remains stable even after ten test cycles. For comparison, they achieved high productivity (3045 L/m^2^·h·bar) and an oil/water emulsion separation efficiency exceeding 99.8% using a 2D MXene/TiO_2_ membrane. One of the primary reasons for incorporating TiO_2_ was to prevent agglomeration of MXene within the interlayer spaces, which positively influenced the photocatalytic properties and permeability of the membrane [195]. However, the authors noted the challenge of evenly distributing ultra-small TiO_2_ particles between MXene layers, as this can lead to aggregation.

**Figure 12 membranes-14-00184-f012:**
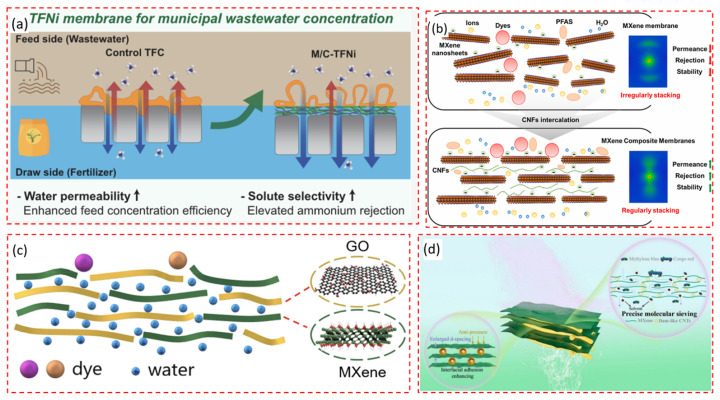
The scheme of water purification by a membrane: (**a**) Thin film composite membrane based on Ti_3_C_2_T_x_/CNT [181]; (**b**) MXene/CeNF [196]; (**c**) MXene/GO [161]; (**d**) MXene/CNT [119].

When using GO in a composite with MXene, attention should be given to the -OH and -COOH functional groups on GO. Incorporating MXene addresses the issue of GO layer aggregation by reducing fluctuations in these groups due to the lower number of oxygen-containing functional groups on MXene nanosheets. This reduction weakens the interactions between these groups and water molecules, thereby increasing water flux. The gaps between adjacent nanosheets in GO/MXene composite membranes create nanoscale channels that allow water to pass through while retaining contaminants. The interlayer spacing (d-spacing) plays a crucial role in determining the water permeability and filtration properties of these composite membranes. In a wet state, the distance between nanosheets is greater than in a dry state due to weakened π-π interactions and the interactions of water molecules with the nanosheets, including hydrogen bonds and electrostatic charges. Increasing the MXene content reduces the number of functional groups in the composite membrane, creating more capillary channels with lower friction coefficients for water passage. This requires a balance between contaminant retention and water flow efficiency [126,184].

Moreover, the binding energy between MXene and contaminants, which depends on the surface groups of MXene and the orientation of the contaminant molecules, can be calculated to evaluate the efficiency of MXene for purifying water from organic contaminants. For instance, calculations by Meng F. et al. [197] demonstrate that urea molecules prefer to align parallel to the surface when adsorbed, regardless of the type of surface group (-OH, -O^−^, or -F). This parallel orientation suggests that urea can penetrate between polymer layers more easily. The most stable adsorption occurs on surfaces with -OH groups, with binding energies of −0.93 eV for parallel orientation and −0.8 eV for vertical orientation. Surfaces with -O^−^ and -F groups follow in stability. Significant charge transfer is observed between urea and -OH surfaces. Such calculations are valuable for predicting the efficiency and selectivity of MXene-based membranes and for understanding the adsorption mechanisms.

In the case of water purification from oils, the hydrophilicity (affinity, presence of pores to promote water passage) and oleophobic properties of composite membranes are crucial, preventing oil passage and facilitating oil-water separation [191,194]. For effective application of such membranes, stability, high water flux, and long lifetime are important considerations.

Membranes based on MXene offer a promising solution for separating oil-in-water emulsions due to their excellent oil removal capabilities and ease of operation [198], including the ability to control the hydrophilicity and hydrophobicity of MXenes [69]. The unique characteristics of MXene nanosheets—such as their numerous surface functional groups, hydrophilicity, and high adsorption capacity—enhance the effectiveness of membranes for oil/water separation across diverse environments, including acidic, alkaline, and saline conditions [199].

One of the main advantages of MXene-modified hydrophilic membranes is their ability to retain a layer of water on the surface due to their roughness, which effectively repels the oil phase. This gives the membranes anti-fouling properties and maintains their functionality in separating organic contaminants, such as oil, from water [200].

However, the low flux of traditional MXene-based membranes due to tight gaps and long interlayer transport remains a problem [201]. Incorporating carbon nanomaterials may offer an effective solution to this challenge, although this field of research is still in its early stages and requires more investigation.

Furthermore, as with other separation membranes, fouling and contamination by organic matter and oil phases can reduce the performance of MXene-based membranes. Various mechanisms have been used to address this problem, including sieving, electrostatic interactions, and oil accumulation [202,203,204]. However, another method, photocatalytic self-cleaning, can be used to extend the life of the MXene membrane. The superhydrophilic and submerged superoleophobic two-dimensional composite MXene membrane developed by Hu J. et al. [205] was used for efficient oil/water separation, demonstrating high permeability and oil repellency for various oil/water emulsions. The membrane exhibits photocatalytic self-cleaning ability due to the β-FeOOH photocatalyst, which provides photo-induced self-cleaning. The optical properties of MXene may also have contributed to this, allowing it to efficiently capture solar radiation and carry out photothermal conversion [69]. The high fouling resistance, low oil adhesion, outstanding reusability, and durability of MXene composite membrane are due to its excellent photoinduced self-cleaning ability.

### 4.4. Dyes

In water purification using MXene-based membranes and carbon-based materials to remove dyes, the molecular weights of the contaminants play a crucial role, as the process essentially involves a ‘sieving’ mechanism [206]. For instance, Congo Red dye, due to its large molecular size, exhibits high rejection rates, reflecting the influence of steric hindrance effects [207]. Additionally, the pH and zeta potential of the membrane impacts the efficiency of water treatment by affecting membrane stability, surface charge, and the ability to attract and retain contaminant molecules [126]. These membrane properties are more easily manipulated when using composites of CNTs and CNFs. For example, the mechanism of water purification using an MXene/CNT composite membrane involves adsorption, molecular sieving, and electrostatic effects that trap dyes between layers. Moreover, π-π electron donor-acceptor interactions and hydrogen bonding play significant roles in adsorption. Studies show that composite MXene/CNT membranes possess more adsorption active sites compared to MXene and CNT separately [127,179].

In membranes based on GO and MXene, nanochannels for water, formed by the addition and oxidation of MXene, play a major role. Increasing the mass content of MXene up to 30% leads to high permeability for pure water while exhibiting high rejection of Rhodamine B [208]. In such membranes, MXene acts as a matrix and provides continuous water channels due to the sequential and uniform distribution of TiO_2_ obtained by the oxidation of MXene, giving it a plate-like structure. It should be noted that while increased water permeability is achieved with higher MXene content, contaminant rejection may decrease. Excessive addition of intercalators can disrupt the internal structure of the GO stacks and reduce separation efficiency, so the optimal mass ratio of MXene to GO must be determined. Nanochannels are effective in separating high molecular weight dyes, preventing them from passing through these channels and allowing the separation of different dyes from a solution [208,209]. This demonstrates the significant potential of the membrane for practical applications in water purification, showing excellent stability under high pressure.

**Table 2 membranes-14-00184-t002:** Summary of the structural features, filtration capabilities, and adsorption capacities of organic and inorganic pollutants by MXene/carbon nanocomposites.

Composite	Pollutant	Tested Concentration	Surface Area (m^2^/g)	Pore Volume (cm^3^/g)	Permeance (L/m^2^·h·bar)	Rejection (%)	Adsorption Capacity (mg/g)	Stability	Source
Ti_3_C_2_T_x_/CNF based on polyvinylpyrrolid	Pb^2+^	10 ppb–10 ppm	n/a	n/a	n/a	89 (in 20 min)	12.7	Removed 77% of Pb^2+^ and 60% of As^3+^ ions in the initial four cycles, but the efficiency notably declined after the fifth cycle	[151]
	As^3+^					81 (in 30 min)	3.3		
Ti_3_C_2_T_x_/CeNF/FeOOH	Sb^+3^	2 mg/L, 12 mg/L	34.15	0.17 (pore size is 6.62 nm)	Retention rate above 94.2%	n/a	19.9	Remained highly reusable after eight cycles, with a minor reduction in removal efficiency for Sb^+3^ and Sb^+5^ by 13.9% and 27.2%, respectively	[176]
	Sb^+5^						18.1		
Ti_3_C_2_T_x_/CNT	Au^+3^	20 ppm	230.6	0.942	437.6	99.8	2093	The rejection rate remained consistent for the first three cycles but decreased to 51.6% over the subsequent six cycles	[179]
Ti_3_C_2_T_x_/CNT in Nylon microfiltration membrane	Crystal violet (CV)	n/a	45.3, membrane thickness is 820 nm)	n/a	1214.3	99.8	n/a	Rejection was 98.3% after 30 h Demonstrated stability in water for 30 h	[210]
	Methyl orange (MO)			n/a	1290.5	95.3			
Ti_3_C_2_T_x_/CNT with polydopamine in α-Al_2_O_3_ substrates	Congo red (CR)	10 mM	n/a	n/a	10.8	>99	n/a	Maintained stability of >97% at pressures ranging from 1 to 5 bar, with a slight decrease in water permeability, and exhibited consistent performance during long-term continuous operation for up to 50 h	[127]
	Rhodamine B (RhB)				11.1	94.9			
	MO				13.2	92.4			
	Na_2_SO_4_	10 ppm			17.4	39.4			
	MgSO_4_				20.9	31.7			
	NaCl				23.5	25.5			
	MgCl_2_				25.9	20.8			
CNT/Ti_3_C_2_T_x_/CNT	NaCl	300–1000 mg/L	25	Pore size is 2–60 nm	rapid average desalination rate 3 mg/g min, 89% retention rate	n/a	34.5	Even after 40 cycles, the cell maintained high desalination capacity	[126]
Ti_3_C_2_T_x_/GO on a mixed cellulose ester membrane	Na_2_SO_4_	5 mmol/L	Thickness is ~ 237 nm	n/a	89.6	60.6	5.1–10.2%	Maintains high stability under high pressure, with rhodamine B rejection remaining over 98% even as the applied pressure increases	[184]
	NaCl					~39.5			
	MgSO_4_					~26			
	MgCl_2_					22.5			
	RhB	10 ppm							
						99.3			
	Methylene blue (MB)					97.6			
	CV					99.1			
	Neutral red (NR)					98.6			
Ti_3_C_2_T_x_/paper membrane	Oil (Sunflower oil, Diesel, Silicone oil, Petroleum oil, Hexane)	1% *v*/*v* oil-in-water	(Effective membrane area is 1.77 cm^2^)	n/a	450	99	Separation efficiency over 99%	No signs of degradation were observed even after eight cycles of operation and washing (demonstrating the membrane’s anti-fouling properties through chemical-free cleaning)	[211]
Ti_3_C_2_T_x_/GO	NaCl	0.1 M	n/a	Pore size is 0.2 µm	25, 6.62, 3.17, 2.14, and 0.79 for water, hexane, toluene, hexane, and IPA; for the NaCl, MgSO4, MR, MnB, RosB, and BB solutions were 2.25, 2.35, 2.1, 0.3, 0.67, and 0.23	<1	n/a	(High removal efficiency of organic dyes with hydrated radii exceeding 0.5 nm)	[212]
	MgSO_4_					5			
	MR	10 mg/L				68			
	MB					99.5			
	Rose Bengal (RosB)					93.5			
	Brilliant blue (BB)					100			
Ti_3_C_2_T_x_/GO	Chrysoidine G	10 mg/L	Thickness is ~550 nm	n/a	6.5	~97	n/a	Remain stable in water over one month (efficiency in >90% separation of dye molecules)	[161]
	NR					99.5			
	MB					99.5			
	CV					~99.5			
	BB					~99.5			
	Humic acid					almost complete removal			
	Bovine serum albumin					almost complete removal			
Ti_3_C_2_T_x_/ GO/Nylon membrane	MO	10 mg/L	Thickness is 140 nm	(Contact angle with water is 34.5°)	Acetone—48.32, Methanol—25.03, Ethanol—10.76, IPA—6.18	98.56	n/a	Stable after 48 h filtration (Rejection: Acetone—1.9%, Methanol—0.5%, Ethanol—0.7%, IPA—1.9%)	[213]
	MB					99.1			
	Eosin					~83			
Ti_3_C_2_T_x_/ CNT cetyltrimethylammonium bromide	Acid orange 7 (AO7)	100 mg/L; adsorbent dosage = 0.5 g/L	56.19	(Pore volume is 0.252 cm^3^/g, pore diameter is 13.77 nm)	n/a	n/a	367.9	The reduction in adsorption efficiency was minimal for AO7 after five cycles, followed by MO and CR. (Na^+^, K^+^, Ca^2+^, Mg^2+^ and Cl^−^, NO_3_^−^, CO_3_^2−^, SO_4_^2−^ had no significant effects on the removal of three dyes)	[214]
	MO						294.2		
	CR						628.5		
Ti_3_C_2_T_x_/ carboxylated CeNF on polydopamine-nylon-66 substrate	NaCl	1.0 g/L	Thicknesses is 129 nm, roughness is 61.6 nm	n/a	Highly permeable	9.7	No adsorption	Exhibited excellent stability, retaining its original morphology after 30 days of immersion in DI water (Separation factor for CR/NaCl: 512.0, for CR/Na_2_SO_4_: 517.2)	[196]
	Na_2_SO_4_					30.7			
	MO	100 mg/L				82.4			
	OG					96.2			
	CR					99.8			
	RB5					96.2			
	perfluoroalkyl substances	1.0 mg/L				94.7			
Ti_3_C_2_T_x_/CNTs coated cotton fabrics (Solar driven interfacial water evaporation system)	Organic dyes (Reactive yellow K-3G, Acid red BG, Disperse navy blue S-2GL), Ions (Na^+^, K^+^, Mg^2+^, Ca^2+^)	~0.7–1030 mg/L for ions, ~200–262 mg/L	(Cotton a plane dimension is 0.5 m × 0.5 m)	n/a	Max wetted radius 30 mm, wetting time 0.324–0.362 s, spreading speed 22.88–24.94 mm/s	Organic dyes removal efficiency > 99%, ions concentration decreased to 0.16–1.12 mg/L	Water adsorption rate 51.14–51.59%/s, water evaporation rate of 1.35 kg m^−2^ h^−1^	Remained almost unchanged after enduring 10 cycles (100 h) of solar evaporation tests in textile wastewater	[215]
Ti_3_C_2_T_x_/CNTs for photocatalytic degradation	RhB	0.01 M dye into 100 mL DI water	n/a	n/a	(Composite is used as catalyst)	Degradation efficiency 75%	n/a	(Enhanced the photo absorption capability and decreased the presence of organic toxic pollutants in wastewater)	[123]
Ti_3_C_2_T_x_/O-multiwalled CNT@polyacrylonitrile	RosB	11 mg/mL	Surface roughness is 14.95 nm	n/a	n/a	99%	n/a	After 21 h rejection was 99%	[216]
	MB					98%			
	CV					100%			
	Janus green (JG) B					99%		After 21 h rejection was 97%	
	air compressor lubricating oil					98%			
p- Ti_3_C_2_T_x_/Single-walled CNT	MB	30–1000 mg/L	1.91	n/a	n/a	97.8%	n/a	Rejection was 95.2% in the fifth cycle of electrosorption	[217]
Ti_3_C_2_T_x_/cellulose acetate	NaCl	2000–4000 ppm	n/a	n/a	256.85\269.02	28.14%	n/a	1.7% weight loss of pristine CA	[218]
	MgCl_2_					40.35%			
	MgSO_4_					56.08%			
	RhB					92.34%			
	MG					98.27%			
	Bovine serum albumin (BSA)					100%			
Ti_3_C_2_T_x_/GO	NaCl	0.2M	Rq is 143 nm Ra is 116 nm	n/a	0.688	99.3%	n/a	Due to dense bonding between nanosheets due to functional groups of components, membrane swelling is suppressed and the permeation rate of ions (K^+^, Na^+^, Li^+^, Al^3+^) is reduced, while the ion sieving characteristics of the membranes were improved by 7–40 times compared to the untreated membrane	[219]
Ti_3_C_2_T_x_/Cellulose Acetate Mixed-Matrix CCAM-10%	Methyl green (MG)	100 mg/L	44.27 m^2^/g	12.83 nm	348.5	96.60%	n/a	There are good antifouling properties: the flux recovery ratio is 67.30% and irreversible fouling ratio at 32.70, as well as improved performance and durability of the membrane for water filtration in cross-flow mode compared to dead-end flow mode.	[135]
	BSA					99.51%			

Achievements in this area include MXene/CeNF membranes, which have been studied for their efficient molecular sieving capabilities [196]. When using cellulose nanofibers for water purification, nanochannel sizes can be controlled due to the strong hydrogen bonds formed between cellulose nanofibers and MXene nanosheets. Lamellar structures and CeNF, which act as ‘threads’ for stitching, contribute to high stability, permeability, and effective molecular separation of dyes. This scheme allows the separation of salts and the purification of wastewater. Differences in the adsorption behaviors of anionic dyes are explained by the number of available adsorption sites and changes in their heterogeneity on the surface. Figure 12b–d illustrate water purification schemes using MXene membranes with different carbon materials.

An alternative purification method is proposed by Wang Y. et al. [215], which uses a solar-powered water surface evaporation system to purify wastewater from dyes. They used a negatively charged Ti_3_C_2_T_x_/positively charged CNT/cotton fabric synthesized by layer-by-layer assembly as a water vapor generator. The synthesized material exhibits exceptional hydrophilicity and stability due to electrostatic and van der Waals interactions and hydrogen bonding. In addition, the material has key properties conducive to water purification, including scalability and low thermal conductivity (0.12 W/m·K). The presence of nanotubes enhances water transport properties, as evidenced by fast wetting times (0.324–0.362 s). The material also absorbs sunlight efficiently (93.5%) due to the optical properties of MXene. These properties make it promising for water purification systems. Water evaporation and condensation allow not only dyes to be separated from water but also salts, while heavy metals remain in the organic phase due to their higher evaporation temperature.

The photodegradation abilities of MXene are particularly effective for water treatment of dyes. For example, an MXene composite membrane used to treat water from oil and dyes has photocatalytic, antifouling, and self-cleaning properties [220]. The authors note that this was achieved by synergistically improving the overall performance of the membrane. Thus, due to membrane separation and adsorption and photodegradation abilities, MXene membrane showed high efficiency and more than 98% dye removal. The use of MXene membrane with these properties for water purification is an environmentally friendly method that allows the decomposition of organic pollutants into less toxic molecules [200,221]. This approach can become an important direction in research oriented towards the development of new water purification technologies.

Therefore, there are different methods and mechanisms for water purification depending on the membrane structure and the type of contaminant. Mechanisms such as photodegradation, adsorption, filtration, and others make membranes based on MXene and carbon nanomaterials a viable option for various water purification applications. However, to fully realize their potential, existing technological and economic barriers need to be overcome.

## 5. Conclusions

MXene-based composites combined with carbon nanomaterials offer a comprehensive array of solutions for purifying water from various contaminants. The unique optical and mechanical properties of MXene, coupled with its high surface area and functional group availability, position this material as a promising candidate for creating effective membranes of diverse compositions. Incorporating additives such as graphene oxide, carbon nanotubes, carbon nanofibers, and cellulose nanofibers further extends the application potential of layered MXene in various water purification methods, as the combination of physicochemical properties of MXene and carbon nanocomposites enhances water purification efficiency and enables the simultaneous retention of multiple types of contaminants.

The synthesis method significantly impacts the structure and adsorption properties of these composites. Carbon materials facilitate the formation of nanochannels and nanobridges and adjust the spacing between MXene layers, which is essential for effective water purification. They also enhance the membrane’s capability to photodegrade organic pollutants. The choice between a membrane or aerogel as the target product depends on the nature of the contaminants—organic or inorganic—and whether the goal is metal recovery or organic molecule rejection.

The primary objective is to develop a stable and practical composite for comprehensive and efficient water purification. This requires further research into synthesizing MXene/carbon nanocomposites through various methods and testing their efficiency in water purification scenarios. The scalability of these technologies is equally important as the issue of water pollution becomes increasingly critical.

While substantial data exist on MXene/carbon nanomaterial hybrid composites, most studies focus on applications in sensors, lithium-ion batteries, and electromagnetic shielding, and some of them on membrane technologies. Researchers working on water purification applications face challenges such as MXene swelling in water. Nevertheless, several studies have demonstrated the potential effectiveness of these composites in water purification, showing high capacity and stability for removing dyes and heavy metals. These promising results for industrial applications necessitate further evaluation based on stability and reusability criteria.

The experimental and theoretical advancements in this field have the potential to provide clean water to millions of people lacking access. Translating laboratory processes to real-life industrial production requires concerted efforts in developing new synthesis methods, improving existing technologies, and conducting large-scale tests under real-world conditions.

## Figures and Tables

**Figure 1 membranes-14-00184-f001:**
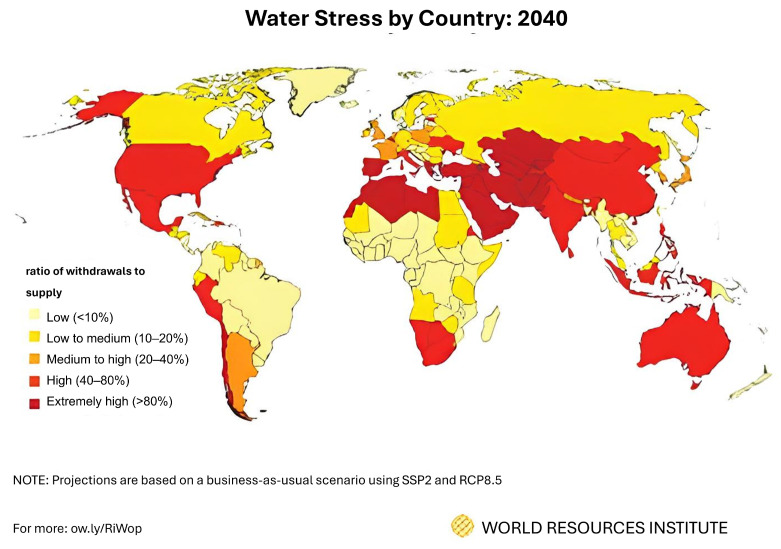
Projections of the water scarcity ranking of countries for the year 2040, according to the Water Resources Institute [7].

**Figure 2 membranes-14-00184-f002:**
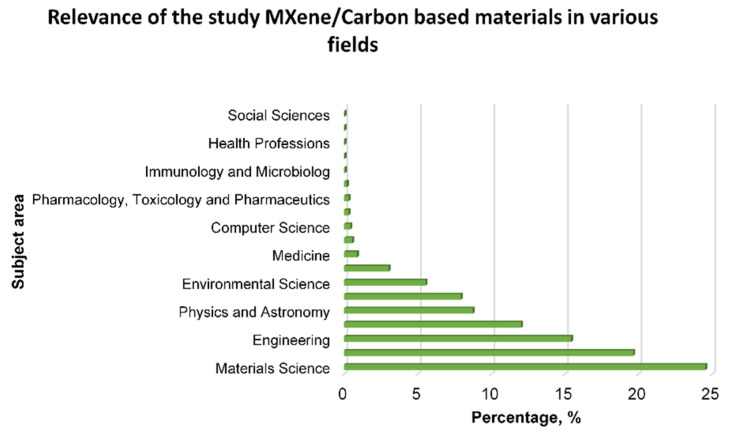
Graph of the relevance of MXene in different fields from 2015 to 2024, based on information from the Scopus database (accessed in May 2024) with keywords: MXene and carbon-based materials.

**Figure 3 membranes-14-00184-f003:**
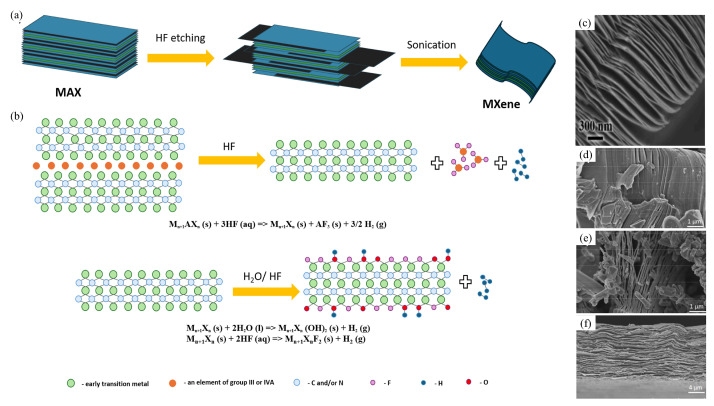
Scheme of MXene synthesis by HF etching and its morphology: (**a**) The scheme of MXene by etching with HF; (**b**) The synthesis and structure diagram of MXene; (**c**) SEM-image of Ti_3_C_2_ [78]; after washing with (**d**) distilled water; (**e**) ethanol; (**f**) MXene synthesized in situ [79].

**Figure 4 membranes-14-00184-f004:**
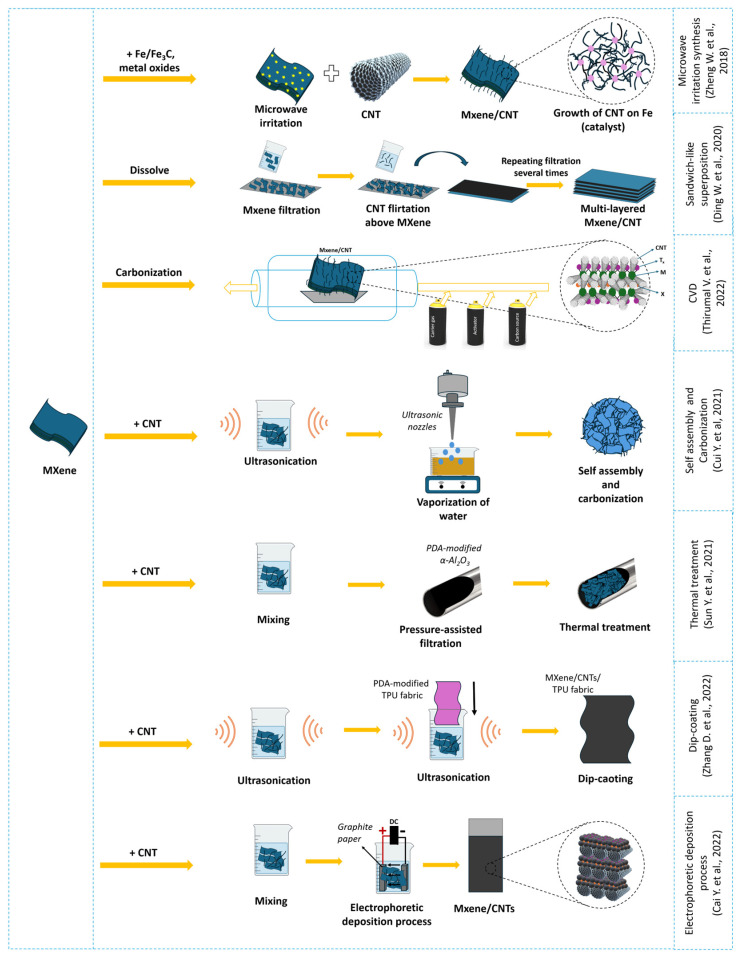
MXene/CNT composite synthesis methods (adapted with permission from references on the right) [121,123,124,125,126,127,128].

**Figure 5 membranes-14-00184-f005:**
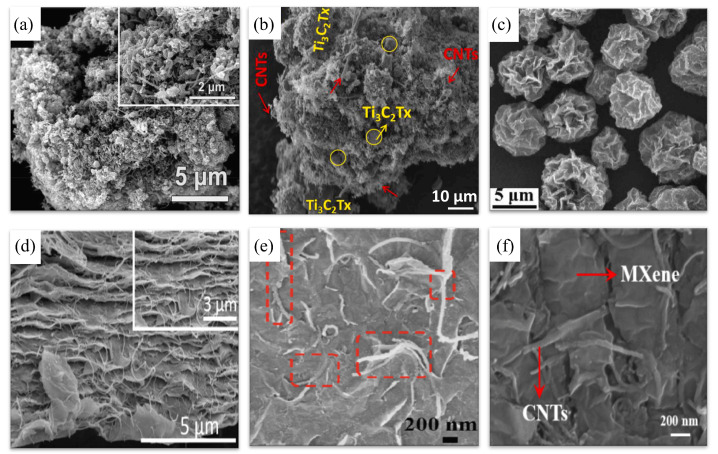
Morphologies of MXene/CNT composites obtained by different methods: (**a**) Microwave irritation synthesis [124]; (**b**) CVD [123]; (**c**) Self-assembly and carbonization [125]; (**d**) Electrophoretic deposition process [126]; (**e**) Thermal treatment [127]; (**f**) Dip-coating [128].

**Figure 8 membranes-14-00184-f008:**
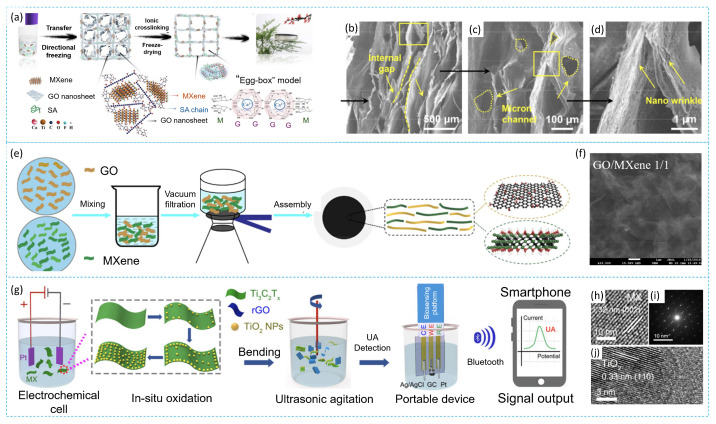
Methods for the synthesis of MXene/GO composites and corresponding results: Directional freeze drying and ion crosslinking for the synthesis of MXene/GO/SA composite (**a**) and SEM images of composite (**b**–**d**) [160]; vacuum filtration of MXene and GO (**e**) and SEM image of composite (**f**) [161]; in situ oxidation of MXene and ultrasonication with GO (**g**), TEM image (**h**) and HRTEM images of composite (**i**,**j**) [162].

**Figure 9 membranes-14-00184-f009:**
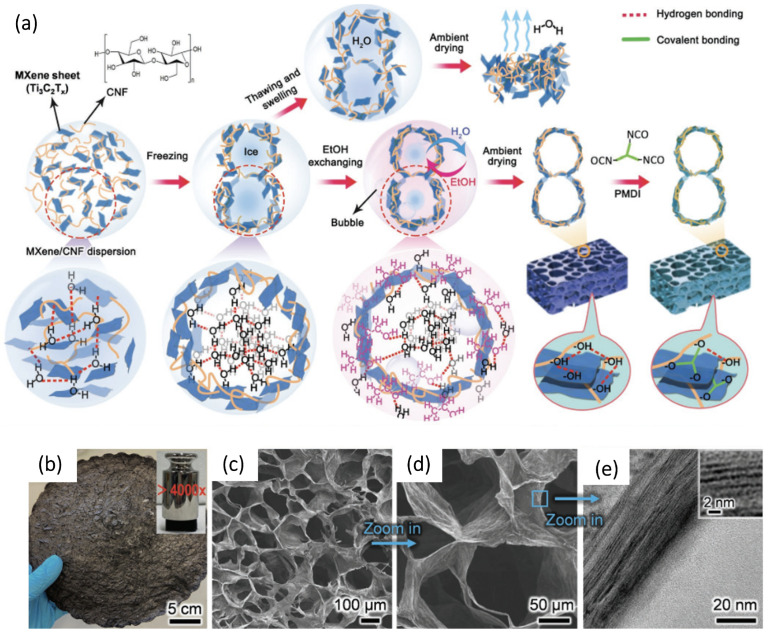
Schematic and morphology of ambient pressure dried MXene based aerogel: (**a**) Schematic of the CeNF assisted ambient pressure drying of MXene based aerogel; (**b**) Photo of MXene based aerogel >4000 times heavier; (**c**–**e**) SEM image of MXene based aerogel [168].

**Figure 10 membranes-14-00184-f010:**
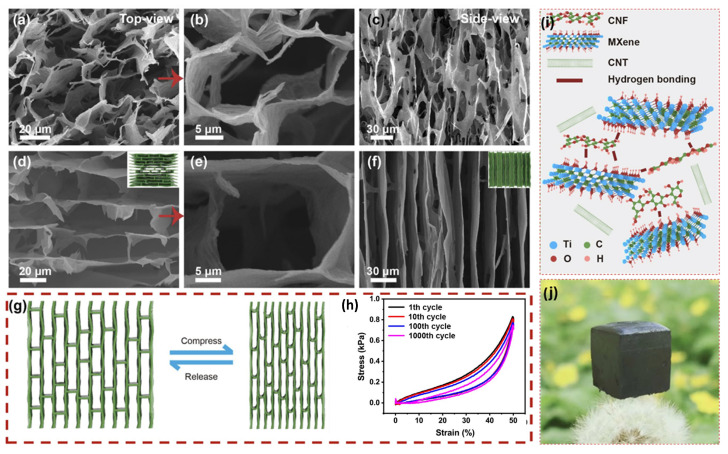
MXene-based aerogel morphology, schematic and aerogel properties: MXene/CeNF/CNT: (**a**,**b**) SEM images of MXene/CNT aerogel with a top view and (**c**) from the side; (**d**,**e**) SEM images of MXene/CeNF/CNT aerogel with a top view and (**f**) from the side; (**g**) Schematic illustration of the compression and release process MXene/CeNF/CNT aerogel; (**h**) Stress/strain curves at 50% elongation for 1000 cycles; (**i**) Schematic illustration of the process of manufacturing aerogels from MXene/CeNF/CNT; (**j**) Photo image of a light aerogel from MXene/CeNF/CNT on the top of a dandelion [169].

**Figure 11 membranes-14-00184-f011:**
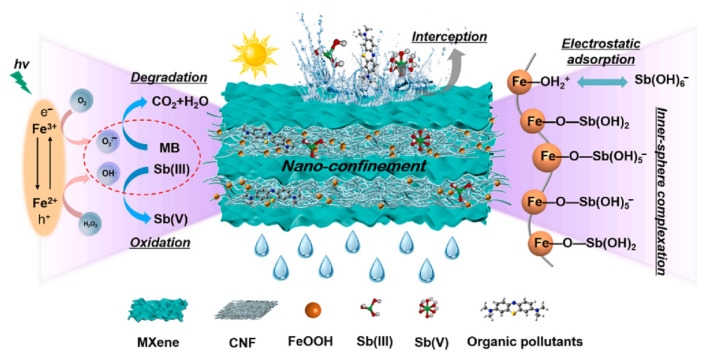
Mechanism of antimony ion removal by self-cleaning membrane Ti_3_C_2_T_x_/CeNF/FeOOH [176].

**Table 1 membranes-14-00184-t001:** MXene/CNT composites obtained by different methods and their features.

Composite	Synthesis Method	Additive	Morphological Features	Adsorption Capabilities	Source
Ti_3_C_2_T_x_/CNTs	Electrostatic self-assembly	basalt fiber-reinforced polymer, epoxy	Uniform thickness; Basalt fibers tightly embedded in matrix composites	Synergistic effect of MXene/CNT/epoxy composite; High stability in alkaline environments	[118]
Ti_3_C_2_T_x_/CNTs	CVD	-	Uniform growth CNTs with a diameter from 40 to 90 nm; Ti_3_C_2_T_x_ layered particles, a common network of Ti_3_C_2_T_x_/CNTs	n/a	[123]
Ti_3_C_2_T_x_/CNTs-cetyltrimethylammonium bromide on nanofiltration membrane	Vacuum-assisted filtration	cetyltrimethylammonium bromide	Membranes with a layered structure had a larger gap between them; The interfacial adhesion force was increased by 6 times compared to the MXene membrane	Membrane has excellent mechanical strength and solvent resistance during molecular sieving; The permeability of pure water increased up to 5 times, with 20.09 L/m^2^·h·bar to 100.89 L/m^2^·h·bar	[119]
Ti_3_C_2_T_x_/CNTs	Electrophoretic deposition	-	CNT provides maximum ion access to ion intercalation sites by increasing the distance between the layers of the MXene nanolayer.	Efficient and fast hybrid capacitive deionization; High fiber hydrophilicity; Specific capacity (178 F/g); Low degreasing resistance; High electrochemical stability (90%); After 1500 cycles and maximum Na^+^ diffusion coefficient; It can provide an energy-efficient desalination process and outstanding desalination stability with a retention rate of 89% after 40 cycles	[126]
Ti_3_C_2_T_x_/functionalized CNTs	Thermal treatment	PDA-modified α-Al_2_O_3_	1D CNTs are well dispersed and embedded in two-dimensional MXene nanoliths; The formation of a homogeneous network and continuous three-dimensional (3D) labyrinthine short mass transfer channels	Improved permeability; Pronounced ability to suppress swelling; Stability	[127]
Dual-phase MoS_2_/Ti_3_C_2_T_x_/CNT	One-step bisolvent solvothermal synthesis technique	1 T enriched-MoS_2_	Triple hybrid structure; Two-phase MoS_2_ (DP-MoS_2_) is formed directly on MXene, while CNTs act as crosslinking between 2D islands; MoO_2_ suppresses oxidation of MXene and rearrangement of 2D layers	Increasing the surface area to 32 m^2^/g	[129]
Ti_3_C_2_/knotted CNTs	CVD	the catalyst Ni–Mn–Al–O	Formation of a three-dimensional network architecture; CNT nodules with a size of 200 ± 20 nm; The average Ti_3_C_2_ flake size is ~250 nm; Ti_3_C_2_/CNT in the form of a sponge	n/a	[122]
Ti_3_C_2_T_x_/CNT/waterborne polyurethane	Sonication	waterborne polyurethane	Free and uniform film with a thickness of 90 µm;	n/a	[130]
Ti_3_C_2_T_x_/ Carboxylated-CNTs microspheres	Self-assembly	-	Layered structure of Ti_3_C_2_T_x_ MXene nanosheets with thickness 1.32 nm; Spherical hierarchical 3D structure of composite with typical shrinkage morphology	BET Surface Area 48.64 m^2^/g; Pore Volume 0.1462 cm^3^/g; Pore Size 24.19 nm	[125]
Ti_3_C_2_T_x_/CNTs	Dip-coating	thermoplastic polyurethane nonwoven fabric	Hypersensitive microcrack structure; Porous fibrous mesh structure	Stability at high temperatures; The synergistic effect of the MXene/CNTs conductive coating	[128]
MXene/sodium lignosulfonate CNT	Self-assembly	sodium lignosulfonate, polyethersulfone substrate pretreated with dopamine	Uniform distribution; Structural integrity	The MB and CR dyes retention efficiency was more than 99% with a permeation flux of 51.6 L/m^2^·h·bar; This membrane shows electrocatalytic efficiency, whereby it degrades various organic dyes (MO, MB, MG, RhB) within 1 h; It has 80% recovery capacity	[134]

## Data Availability

Not applicable.

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
