# Peer review of "MXene/Carbon Nanocomposites for Water Treatment"

_membranes, 2024, doi:10.3390/membranes14090184_

Round 1
Reviewer 1 Report
Comments and Suggestions for Authors
This paper considers the potential of already discovered MXenes in combination with carbon nanomaterials to solve the growing problem of water pollution providing the industry with suitable membranes for water purification.
The review provides all the necessary background information on the production and use of the materials as well as on the creation of composites. The advantages and challenges are discussed in detail with focus on water purification and membrane formation.
Very well done and I recommend the publication.
Author Response
Reviewer 1
Comment: This paper considers the potential of already discovered MXenes in combination with carbon nanomaterials to solve the growing problem of water pollution providing the industry with suitable membranes for water purification. The review provides all the necessary background information on the production and use of the materials as well as on the creation of composites. The advantages and challenges are discussed in detail with focus on water purification and membrane formation. Very well done and I recommend the publication.
Response: Thank you very much for your positive and encouraging feedback. We are delighted to hear that you found our manuscript well-done and suitable for publication. We deeply value your opinion and the time and effort you invested in the review process. We are sincerely grateful for your support.
Reviewer 2 Report
Comments and Suggestions for Authors
In this manuscript, the authors discuss the synthesis methods of MXene and its composites, their structure, properties, and mechanisms of water purification from various pollutants evaluates. MXene combined with carbon nanomaterials can provide effective water treatment solutions. However, there are some concerns.
1. Are there any MXene used in the water treatment field with in-situ synthesis, hydrothermal synthesis or fluorine-free synthesis?
2. The references corresponding to each method in Figure 4 and Figure 6 should be clearly labelled.
3. The use of "adsorption capacity" in Table 1 is not appropriate.
4. It would be better if more articles in the field of water treatment cited in Chapter 3.
5. On page 18, line 620, it is necessary to explain the mechanism of electrostatic adsorption of antimony ions by Ti3C2Tx/CNF/FeOOH membrane.
6. In Chapter 4.2, The mechanism of salts removal needs to be explained more detailedly.
7. There are some details in Table 2 that need to be corrected. The unit of rejection is not consistent. The “18 m” is a clerical error. On page 18, the rejection rate of eosin by Ti3C2Tx /GO/Nylon membrane is written wrongly, which is written as 108.3.
Comments on the Quality of English LanguageMinor editing of English language required
Author Response
Reviewer 2
In this manuscript, the authors discuss the synthesis methods of MXene and its composites, their structure, properties, and mechanisms of water purification from various pollutants evaluates. MXene combined with carbon nanomaterials can provide effective water treatment solutions. However, there are some concerns.
Comment #1: Are there any MXene used in the water treatment field with in-situ synthesis, hydrothermal synthesis or fluorine-free synthesis?
Response #1: Thank you for pointing this out. Information on MXene used in water treatment with in-situ synthesis, hydrothermal synthesis, and fluorine-free synthesis has been added in lines 636-644.
Comment #2: The references corresponding to each method in Figure 4 and Figure 6 should be clearly labelled.
Response #2: Thank you for pointing that out. However, we would like to clarify that the figures were plotted and the schemes in the figures were designed by the authors, which is why there are no references included.
Comment #3: The use of "adsorption capacity" in Table 1 is not appropriate.
Response #3: Thank you for your observation. We appreciate your attention to detail. Upon review, we found that Table 1 does not mention "adsorption capacity" but rather "adsorption capabilities".
Comment #4: It would be better if more articles in the field of water treatment cited in Chapter 3.
Response #4: Thank you for your valuable suggestion. We have incorporated additional references in Chapter 3, now numbered [119], [132], [154], and [160].
Comment #5: On page 18, line 620, it is necessary to explain the mechanism of adsorption of antimony ions by Ti3C2Tx/CNF/FeOOH membrane.
Response #5: We appreciate your comments. We have added an explanation in the review on lines 667-675.
Comment #6: In Chapter 4.2, The mechanism of salts removal needs to be explained more detailedly.
Response #6: Thank you for your comment. We have expanded Chapter 4.2 to provide a more detailed explanation of the mechanism of salts removal. The new information, located on lines 763-768, 772-779, and 785-797, delves into the processes and interactions involved in the removal of salts, offering a clearer and more comprehensive understanding of the topic.
Comment #7: There are some details in Table 2 that need to be corrected. The unit of rejection is not consistent. The “18 m” is a clerical error. On page 18, the rejection rate of eosin by Ti3C2Tx /GO/Nylon membrane is written wrongly, which is written as 108.3.
Response #7: Thank you for noticing that. Adjustments have been made in response to the comment.
Minor editing of English language required
We have made the necessary edits to the English language in the manuscript.
We sincerely appreciate your review and the invaluable feedback you provided. Your comments and suggestions have greatly contributed to enhancing the quality of our manuscript. We are deeply grateful for your time, effort, and insightful input.
Reviewer 3 Report
Comments and Suggestions for Authors
In this work, studies on MXene/carbon nanomaterials for water treatment are summarized and discussed. The topic may be a little out of scoup of this special issue as the materials are not only made for membranes. Other suggestions are as the following:
1. The title of this review is a litttle misleading for the readers. If the authors hope to focus on the MXene and carbon based nanocomposite materials, the title may be changed to "MXene/Carbon Nanocomposites for Water Treatment".
2. Some important references are missing, such as Journal of Membrane Science 2024, 700, 122691; Science Bulletin 2024, 69,125-140; Chemical Engineering Journal 2023, 474, 145579.
3. In this review, the preformance of the nanocomposites were classified by the components to be removed. This might not be very resonable. The authors are suggested to classify by the functions like adsorbents, membranes, and etc.
Comments on the Quality of English LanguageThe English for the manuscript is satisfactory for publication.
Author Response
Reviewer 3
In this work, studies on MXene/carbon nanomaterials for water treatment are summarized and discussed. The topic may be a little out of scoup of this special issue as the materials are not only made for membranes. Other suggestions are as the following:
Comment #1: The title of this review is a litttle misleading for the readers. If the authors hope to focus on the MXene and carbon based nanocomposite materials, the title may be changed to "MXene/Carbon Nanocomposites for Water Treatment".
Response #1: We agree with your opinion, thank you very much. We have changed the title of the review.
Comment #2: Some important references are missing, such as Journal of Membrane Science 2024, 700, 122691; Science Bulletin 2024, 69,125-140; Chemical Engineering Journal 2023, 474, 145579.
Response #2: Thank you very much for the helpful articles. We have cited them in our review with reference numbers [56] (Science Bulletin), [132] (Chemical Engineering Journal), and [175] (Journal of Membrane Science).
Comment #3: In this review, the preformance of the nanocomposites were classified by the components to be removed. This might not be very resonable. The authors are suggested to classify by the functions like adsorbents, membranes, and etc.
Response #3: Thank you very much for such a critical comment. In explaining our point of view, since the structure of carbon nanocomposites affects the structure of the composite with MXene, which consequently determines its adsorption characteristics and the mechanism of water purification from different types of pollutants, we decided that classification by type of nanocomposites would be acceptable and understandable. Certainly, they can be classified as you suggest (adsorbent, membrane, etc.). However, we would also like to point out the differences between MXene composites with different carbon nanomaterials. However, we appreciate your review and that we helped us revise the review title and improve it.
We also greatly appreciate your review and the assistance you provided in revising and improving the title of our manuscript. Your help has been instrumental in enhancing the clarity and impact of our work.
Reviewer 4 Report
Comments and Suggestions for Authors
The review of the paper by Keneshbekova et al. The paper is quite good, containing a lot of interesting data and information. Thank you for your work.
However, I cannot recommend this paper for publication in Membranes because it is not primarily focused on membranes. Instead, this paper would be more suitable for submission to another journal, such as Materials.
Additionally, there is a lack of citations to relevant papers, also published in Membranes in recent years, that should be included in this review.
Author Response
Reviewer 4
Comment #1: The review of the paper by Keneshbekova et al. The paper is quite good, containing a lot of interesting data and information. Thank you for your work. However, I cannot recommend this paper for publication in Membranes because it is not primarily focused on membranes. Instead, this paper would be more suitable for submission to another journal, such as Materials. Additionally, there is a lack of citations to relevant papers, also published in Membranes in recent years, that should be included in this review.
Response #1: Thank you so much for your comment. We have supplemented the references of relevant articles that confirm what was written in our paper ([37], [42], [56], [119], [132], [154], [168], [169], and [175].), including those published by the journal Membranes ([160], and those previously present [191] and [192]).
We also extend our heartfelt thanks for your review and for assisting us in enhancing our article. Your feedback has been invaluable in improving the overall quality of our work.
Reviewer 5 Report
Comments and Suggestions for Authors
Review comments are attached.

Comments on the Quality of English LanguageAuthor Response
Reviewer 5
This manuscript comprehensively reviews the synthesis methods, characteristics, and water treatment applications of MXene/carbon nanomaterials. Additionally, it addresses the limitations of MXene/carbon nanomaterial composites for practical applications and proposes future directions for enhancing their water treatment performance. The manuscript is well-structured, with clearly organized sections and well-drawn figures that summarize the content. It thoroughly discusses critical information, such as the characteristics of nanomaterials, their mechanisms in pollutant removal, and the synergistic effects of the nanocomposites. However, to further improve the quality of the work and make it suitable for publication, the author should consider the following suggestions and questions. Questions and suggestions:
Comments #1: The primary concern is that this review manuscript focuses mainly on MXene/carbon nanomaterials for water treatment, rather than on MXene/carbon nanocomposite membranes. Although the applications of this nanomaterial in membrane development are mentioned, it raises the question of whether the manuscript fits the scope of this journal. If the author team intends to submit to a journal specializing in membranes, they should place more emphasis on reviewing MXene/carbon membranes.
Response #1: Thank you for your constructive comments. In response to your feedback, we have expanded our discussion on MXene/carbon nanocomposite membranes, incorporating additional details as referenced in [119] (Table 1 and lines 315-321), [132] (lines 411-413), and [168] (lines 772-779). We have also included information on other membranes with MXene, as noted in [160] (lines 688-691) and [175] (lines 815-818) for comparison. Also, let us note that the details of MXene/carbon nanocomposite membranes are summarized in Table 2.
Comments #2: Figure 2, the detailed information regarding the resource of data in this figure should be given. For example, are there any keywords or conditions used in the searching process in the Scopus database?
Response #2: Thank you for pointing that out. The information has been added to the review (line 64-65).
Comments #3: Page 3, “…carbon nanomaterials such as graphene oxide, … are of particular interest due to their unique adsorption properties and environmental neutrality”, more explanations are required on the “unique adsorption properties” and the “environmental neutrality”.
Response #3: Thanks to your comment, we were able to supplement this information. Explanations are provided in lines 70-75.
Comments #4: Figure 3f is neither marked in figure title nor discussed in the text.
Response #4: Thank you for noticing the omission. A mark and a citation (line 163) have been added to the text.
Comments #5: Line 227-228, “These composites have new possibilities for creating materials with diverse functions, including sensing…”, please revise this sentence as it is not precise. For example, sensing, catalysis, and water desalination are not “functions”.
Response #5: Indeed, the sentence sounds inaccurate, thanks for noticing. It has been corrected.
Comments #6: The synthetic effects of MXene/carbon nanocomposites, the related mechanisms, and the advantages should be comprehensively discussed.
Response #6: Assuming that there was a synergistic effect of MXene/carbon nanocomposites, we have supplemented this information in lines 316-322 and 646-654. Thus, thanks to you, we were able to bring this point out better. That's valuable to us.
Comments #7: Line 251-252, “Among the various methods, the self-assembly method stands out for its accessibility and practicality.”, more explanation is required regarding this conclusion.
Response #7: Thank you for your comment. We have added an explanation, and better disclosed this point (line 264-276).
Comments #8: Figure 8 is not clear. A high-resolution figure is required
Response #8: Thank you for noticing that. The quality of Figure 8 has been improved and modified in the review.
We are deeply grateful for your review, which means a great deal to us. Your thoughtful feedback has been instrumental in improving the quality of our manuscript. Thank you for your invaluable contribution.
Round 2
Reviewer 2 Report
Comments and Suggestions for Authors
Accept in present form
Author Response
We are grateful for the time and effort you have put into your review of our work.
Reviewer 4 Report
Comments and Suggestions for Authors
Thank you for your work. I did not change my opinion. I think the paper is good, but it should be published in another journal. It is not relevant for Membranes.
Author Response
We greatly appreciate your valuable comments. Accordingly, we have tried to improve our article by adding more discussion about membranes, and we hope that after all the corrections, our article has become more meaningful and clear.
Reviewer 5 Report
Comments and Suggestions for Authors
No
Comments on the Quality of English LanguageManuscripts should be carefully proofread to avoid English errors.
Author Response
We would like to express our sincere gratitude for your time and effort spent reviewing our work. Your comments and recommendations have been extremely valuable and have significantly helped us improve the quality of the research. The work has been carefully proofread, and the English has been enhanced.
Round 3
Reviewer 4 Report
Comments and Suggestions for Authors
I did not change my opinion. The paper is good, but it is not proper for Membrane journal.
Author Response
We would like to extend our sincere gratitude for your thorough and constructive review of our manuscript. Your insightful comments and valuable suggestions have significantly improved the quality and clarity of our work. We deeply appreciate the time and effort you invested in providing such detailed feedback, which has helped us to refine and strengthen our paper.
Reviewer 5 Report
Comments and Suggestions for Authors
NO
Author Response

(The authors gave the same response as above.)
